

# How is particulate organic carbon transported through the river-fed Congo Submarine Canyon to the deep-sea?

Sophie Hage[1*], Megan L. Baker[2], Nathalie Babonneau[1], Guillaume Soulet[1], Bernard Dennielou[1], Ricardo Silva Jacinto[1], Robert G. Hilton[3], Valier Galy[4], François Baudin[5], Christophe Rabouille[6], Clément Vic[7], Sefa Sahin[8], Sanem Açikalin[8], Peter J. Talling[2]

[1]Geo-Ocean, UMR6538, Univ Brest, Ifremer, CNRS, Plouzané, France
[2]Departments of Geography and Earth Sciences, Durham University, UK
[3]Department of Earth Sciences, University of Oxford, UK
[4]Department of Marine Chemistry and Geochemistry, Woods Hole Oceanographic Institution, MA, USA
[5]ISTeP, UMR 7193, Sorbonne Université, CNRS, Paris, France
[6]LSCE, UMR 8212, CEA-CNRS-UVSQ, IPSL and Université Paris-Saclay, Gif-sur-Yvette, France
[7]Univ Brest, CNRS, Ifremer, IRD, Laboratoire d'Océanographie Physique et Spatiale (LOPS), IUEM, Plouzané, France
[8]School of Natural and Environmental Sciences, Newcastle University, UK
*Correspondence to*: Sophie Hage (sophie.hage@univ-brest.fr)

## Abstract

The transfer of carbon from land to the near-coastal ocean is increasingly being recognized in global carbon budgets. However, a more direct transfer of terrestrial carbon to the deep-sea is comparatively overlooked. Among systems that connect coastal to deep-sea environments, the Congo Submarine Canyon is of particular interest since the canyon head starts 30 km into the Congo River estuary, which delivers ~7% of the total organic carbon from the world's rivers. However, carbon and sediment transport mechanisms that operate in the Congo Canyon, and submarine canyons more globally, are poorly constrained compared to rivers because monitoring of deep-sea canyons remains challenging. Using a novel array of acoustic instruments, sediment traps and cores, this study seeks to understand the hydrodynamic processes that control delivery of particulate organic carbon via the Congo Submarine Canyon to the deep-sea. We show that particulate organic carbon transport in the canyon-axis is modulated by two processes. First, we observe periods where the canyon dynamics are dominated by tides, which induce a background oscillatory flow (speeds of up to 0.15 m/s) through the water column, keeping muds in suspension, with a net upslope transport direction. Second, fast-moving (up to 8 m/s) turbidity currents occur for 35 % of the time during monitoring periods and transport both muddy and sandy particulate organic carbon at an estimated transit flux that is more than ten times the flux induced by tides. Remarkably, organic carbon transported and deposited in the submarine canyon has a similar isotopic composition to organic carbon in the Congo River, and in the deep-sea fan at 5 km of water depth. Episodic turbidity currents, together with background tidal currents thus promote efficient transfer of river-derived particulate organic carbon in the Congo Submarine Fan, leading to some of the highest terrestrial carbon preservation rates observed in marine sediments globally.



## 1 Introduction

Earth's carbon is exchanged between land, ocean and atmosphere reservoirs, regulating life and climate over daily to geological timescales. The land and ocean carbon reservoirs are usually quantified separately in global carbon cycle models, and viewed in terms of their carbon exchanges with the atmosphere (Ciais et al., 2013, Friedlingstein et al., 2022). However, an emerging view of the global carbon cycle is now taking into account the role of land to ocean transport of carbon, which is a potential source or sink of atmospheric $CO_2$ (Galy et al., 2015, Regnier et al., 2013, 2021). It has long been thought that most (90%) terrestrial sediment and particulate carbon reaching the ocean stay on deltas and continental shelves (Berner, 1989, Burdige, 2007, Tegler et al., in press), particularly during sea level highstands (Posamentier and Kolla, 2003). However, an increasing number of studies have revealed that large volumes of terrestrial and shelf sediment can be transferred via submarine canyons to the deep-sea, where it is buried in depositional systems called deep-sea fans (e.g., Canals et al., 2006, Galy et al., 2007, Kao et al. 2014, Rabouille et al., 2019, Talling et al. 2024). Despite their importance for carbon cycling, submarine canyons remain poorly investigated compared to river systems on land. This is because particulate transport in submarine canyons includes seafloor sediment-flows called turbidity currents, which are episodic, challenging to measure, and can damage monitoring instruments (Khripounoff et al., 2003, Sumner et al., 2014; Hughes Clarke, 2016, Talling et al., 2023). Transport processes in submarine canyons are thus poorly constrained, with understanding still mostly based on interpretation of seabed sediments and modelling (Bouma, 1962, Symons et al., 2017 Cartigny et al., 2013, Miramontes et al., 2020, Ge et al., 2022).

A major step-forward in understanding sediment transfer in submarine canyons was made possible in the past two decades through advances in technology that enabled direct measurements of sediment transport processes in canyons. Observations show that turbidity currents are the main transport process for moving material down-canyon (e.g., Congo, Monterey, Whittard, Var, Capbreton Canyons; Khripounoff et al., 2003, 2012, Azpiroz-Zabala et al., 2017, Paull et al., 2018, Maier et al., 2019, Simmons et al., 2020, Guiastrennec-Faugas et al., 2021; Talling et al., 2022, 2023). These flows last from a few minutes to up to a few days and sometimes travel at high speed (e.g., 19 m/s; Heezen and Ewing, 1952). However, other more continuous processes such as internal tides, cascading or upwellings also occur in submarine canyons (Shepard et al., 1979). These oscillating currents, with directions that reverse up-canyon to down-canyon, typically have speeds of up to 0.3 m/s. But in a few locations, oscillations due to tides and winds can sometimes reach moderate velocities (up to 1 m/s; e.g., Monterey, Capbreton, Logan, Whittard and Cassidaigne Canyons; Mulder et al., 2012, Maier et al., 2019, Li et al., 2019, Heijnen et al., 2023, Brun et al., 2023). Yet, the way background oscillatory flows and episodic turbidity currents





interact and combine to transport sediment and organic carbon to the deep-sea is poorly constrained. This lack of understanding inhibits an accurate assessment of sediment and carbon transport from continental shelves to the deep-sea.

In this study, we use direct observations of transport processes and sampling of particulate organic carbon from the Congo Submarine Canyon. The latter provides a unique opportunity to test how particulate organic carbon transport by rivers on land continues into the ocean because the canyon head starts within the Congo River estuary (Fig. 1). This connection is important because the Congo River ranks fifth in terms of global particulate organic carbon export to the ocean (Coynel et al., 2005). The Congo Submarine Canyon continues as a less deeply incised

submarine channel, beyond which is an area termed a lobe, where flows become unconfined and very rapid sedimentation occurs (Babonneau et al., 2002; Dennielou et al., 2017). The lobe is at 5 km water depth and constitutes a sink of organic carbon, with estimated burial rates of 0.42 Mt C/yr, equalling 19% of the total organic carbon buried annually in the South Atlantic Ocean at >3000 m water depth (Rabouille et al., 2019, Mollenhauer et al., 2004). The submarine canyon, channel and lobe together form the Congo Submarine Fan. The Congo

Submarine Fan has received widespread attention in the past two decades, (e.g., Savoye et al., 2000, Babonneau et al., 2003, 2010, Khripounoff et al., 2003, Rabouille et al., 2009, 2017a,b, 2019, Baudin et al., 2010, 2017, 2020, Stetten et al., 2015, Dennielou et al., 2017, Talling et al., 2022, Pope et al., 2022). A range of hydrodynamic processes has been directly observed (Khripounoff et al., 2003, Azpiroz-Zabala et al., 2017, Talling et al. 2022). For example, both occasional and fast (up to 3.5 m/s) turbidity currents, and continuous and slow (up to 10 cm/s)

internal tides were detected in 2004 in the distal channel, which is 1000 km downstream of the river estuary (Vangrieshem et al., 2009). Measurements in 2010 and 2013 showed that turbidity currents in the proximal canyon last for up to a week, occur for ~30% of the time during monitoring periods, and reach speeds of 2-3 m/s (Azpiroz-Zabala et al., 2017, Simmons et al., 2020). More recently, fourteen turbidity currents were detected in the upper Congo Canyon including one flow which travelled >1,130 km while accelerating from 5.2 to 8.0 m/s, breaking

seabed communication cables and moorings on 14 January 2020 (Talling et al. 2022). Despite numerous studies of the Congo Submarine Fan, important questions remain related to hydrodynamic processes and associated carbon transfer in the canyon towards the deep-sea.

This paper aims to further our understanding of carbon transport in the Congo Submarine Canyon by linking for

the first time detailed organic carbon measurements with direct observations and sampling of flows in the canyon obtained in 2019-2020. The following three questions will be discussed. First, what are the hydrodynamic processes





controlling the transport of particulate organic carbon in the Congo Submarine Canyon? In particular, what is the relative importance of oscillatory slow currents (e.g. tides), and faster but more episodic turbidity currents. Second, how does the composition and age of organic carbon transiting through the canyon, and deposited on the deep-sea lobe, compare to that of particulate organic carbon in the Congo River? This comparison allows us to discuss the efficiency of terrestrial organic carbon transport to the deep-sea reservoir. Third, how do hydrodynamic processes and associated carbon transport in the Congo Submarine Canyon compare with submarine canyons in different settings? This leads us to briefly discuss the implications for the redistribution of carbon between two globally important carbon reservoirs (i.e., terrestrial biosphere and ocean sediments).

## 2 The Congo River and deep-sea fan system

### 2.1 The Congo River and connected submarine canyon

The Congo River is the second largest river in the World in terms of drainage area ($3.7 \times 10^6$ km²). The hydrological regime of the lower river (Brazzaville-Kinshasa gauging station) is equatorial with limited annual fluctuations and an average annual discharge of 40,000 m³/s for the years 1903-1995 (Bricquet et al., 1995). The Congo River mouth forms an estuary under a microtidal regime (up to 1.9 m tidal range; Vallaeys et al., 2021). A submarine canyon starts thirty kilometres into the river estuary from the coast. This canyon is 300 km long, 10 km wide, up to 800 m deep, and is characterized by multiple terrace levels (Fig. 1A; Heezen et al., 1964). The canyon then evolves into a 600 km long channel overhung by levees and terminates to distal depositional lobes at 5 km of water depth (Savoye et al., 2000, Babonneau et al., 2002). Downslope turbidity currents were directly observed in the channel (Khripounoff et al., 2003, Vangriesheim et al., 2009) and more recently in the canyon (Azpiroz-Zabala et al., 2017, Simmons et al., 2020; Talling et al., 2022) using moored current meters (Fig. 2). Fourteen turbidity currents were measured over four months in 2019 and 2020 in the proximal canyon, at a site 200 km downstream of the Congo River mouth (Talling et al., 2022). One of these flows travelled more than 1,100 km down the canyon and channel system, reaching transit velocities of 8 m/s and damaging instrumented moorings and seabed telecommunication cables. An estuarine interplay of river floods combined with spring-tide, are suggested as the triggering mechanism for those flows (Talling et al., 2022) yet more direct observations are needed to test this hypothesis.

### 2.2 Past studies of particulate organic carbon in the Congo River and deep-sea fan system

The Congo River exports ca. 2 Mt C/yr of particulate organic carbon annually, ranking fifth in terms of particulate organic carbon export by rivers globally (Coynel et al. 2005). The composition of this riverine carbon is dominated



by C3 plant debris derived from the rainforest vegetation and by pre-aged soil from the Cuvette Congolaise swamp forest (Spencer et al., 2012, Hemingway et al., 2017), with additional minor contributions from freshwater phytoplankton and C4 plants-derived organic matter. The contribution of fresh C3 vegetation was shown to increase with elevated river discharge (Hemingway et al., 2017). River suspended sediment samples measured for organic carbon composition show wide ranges of total organic carbon contents (TOC from 5.7 to 11.8 %), carbon stable isotope ($\delta^{13}$C from -28.8 ‰ to -24.6 ‰) and radiocarbon isotope compositions ($\Delta^{14}$C from -309‰ in the fine particulate organic carbon fraction to 95.7‰ in the coarse particulate organic carbon fraction; Spencer et al., 2012, Hemingway et al., 2017).

Between 33 and 69 % of the Congo River particulate organic carbon export are estimated to be deposited in the Congo Submarine Fan including the canyon, channel, levees and distal lobe environments based on geochemical measurements from surficial (20 cm deep) sediment (Rabouille et al., 2019, Stetten et al., 2015, Baudin et al., 2010). Longer piston sediment cores (up to 10 m long) from the channel and levees at 4 km water depth, reveal that 70 to 80 % of the organic matter is terrestrial and derived from the river (Baudin et al., 2010). Piston sediment cores were recently retrieved from the canyon, covering a variety of grain sizes with TOC contents ranging from 0.15 to 11.3 %, and $\delta^{13}$C from -28.5 to -26.2 ‰ that are similar to samples from the Congo River (Baker et al., in review). The only marine signature for organic carbon composition in the Congo Submarine Fan was found in a sediment trap located 45 m above seabed at 4779 m water depth in the distal channel to lobe transition zone (Fig.1 A, Baudin et al., 2017). TOC contents in this sediment trap vary between 3.9 and 5.6%, whereas $\delta^{13}$C range from -24.4 to -22.7 ‰ (Baudin et al., 2017).





**Figure 1: The Congo River and submarine system. A.** Map of Congo River and bathymetry of its connected submarine canyon and channel terminating onto distal lobe. Location of distal channel multi-cores collected for this study, and location of the cores and sediment trap deployed 45 m above seabed ([1]Baudin et al., 2017). **B.** Location of the moored instruments (M9 mooring in Talling et al., 2022) and of the sediment cores used in this study in the Congo submarine canyon at 2190 m water depth. **C.** Cross-section across the submarine canyon showing the locations of the acoustic Doppler current profiler (ADCP), sediment trap and multi-cores.



## 3 Methods

### 3.1 Oceanographic observations

In total, nine acoustic Doppler current profilers (ADCP) were deployed on moorings in the Congo submarine canyon and channel in September-October 2019 during the JC187 cruise aboard the James Cook research vessel

(Talling et al., 2022). Two moorings surfaced in October 2019 due to turbidity currents in the upper canyon whereas the last seven moorings surfaced by January 2020 due to a powerful turbidity current. The latter ran out to the end of the submarine channel (>1200 km off the river mouth), and accelerated from 4.5 to 8 m/s, breaking two telecommunication cables deployed on the seabed (WACS and SAT-3; Talling et al., 2022). In this study we focus on data recovered from one mooring (M9 in Talling et al., 2022; Fig. 1) that included one ADCP and a sediment

trap. Our studied mooring (M9) is located 200 km off the river mouth, at 2190 m water depth in the canyon thalweg (Fig. 1A). At this location, the canyon-thalweg is incised by 500 m, and has two terrace levels 200 m and 250 m above the thalweg (Fig. 1C). The 600 kHz ADCP was downward-looking, and located 32 m above the seabed. The instrument recorded velocity data in three directions (eastward, northward and vertical) at 11 second intervals, with a vertical cell size of 0.75 m. All three velocity directions were combined to generate a velocity magnitude ($V_{mag}$)

matrix (Fig. 2). Elevated values of $V_{mag}$ allow turbidity current events to be identified. In total, fourteen turbidity currents were detected in the Congo Canyon between 15[th] September 2019 and 14[th] January 2020 (Talling et al., 2022). In this study, we focus on the first eight turbidity currents out of the fourteen events reported in Talling et al. (2022). These eight flows occurred between 15[th] September and 10[th] December 2019 and were less powerful than the 14[th] January event. The last six flows are not further considered here because they were not identified in

the sediment trap due to disturbance in the trap top (Fig. A1).

To further analyse the motion of water and particles in the canyon between turbidity current events, the ADCP's eastward and northward velocities were turned into $V_{along}$ (velocity along canyon) and $V_{cross}$ (velocity across canyon) using the direction of the canyon axis at our mooring location (Fig. 3). Hodographs were created for two $V_{along}$ arrays 5 m and 30 m above canyon floor. Hodographs are progressive vector diagrams showing how velocity

vectors change with time from a fixed, common origin (Hamilton, 1847), with our M9 mooring location chosen as the origin. The varying directions and magnitudes in $V_{along}$ expressed in these two hodographs show the main trajectory derived from Eulerian residual currents at the M9 location (Fig. 4). Finally, a frequency analysis was carried out on the ADCP velocity data to identify the main period peaks in velocity oscillations to be correlated with known hydrodynamic processes such as tides (Fig. 3). Power spectral densities were computed using the



Welch method from the data binned on a regular 10-min grid, with 2048-bin segments and a half-segment overlap (Welch 1967).

We also present data from one Conductivity Temperature Depth profiler (CTD) cast, which was collected at the M9 location on 19 September 2019, when there was no turbidity current event (Fig. 5). The instrument is a WetLab Eco BBRTS Scattering meter, which provided salinity (in PSU), oxygen (in µmol/kg) temperature (in °C) and turbidity (in m$^{-1}$ sr$^{-1}$) profiles from water surface to canyon floor at 2190 m water depth.

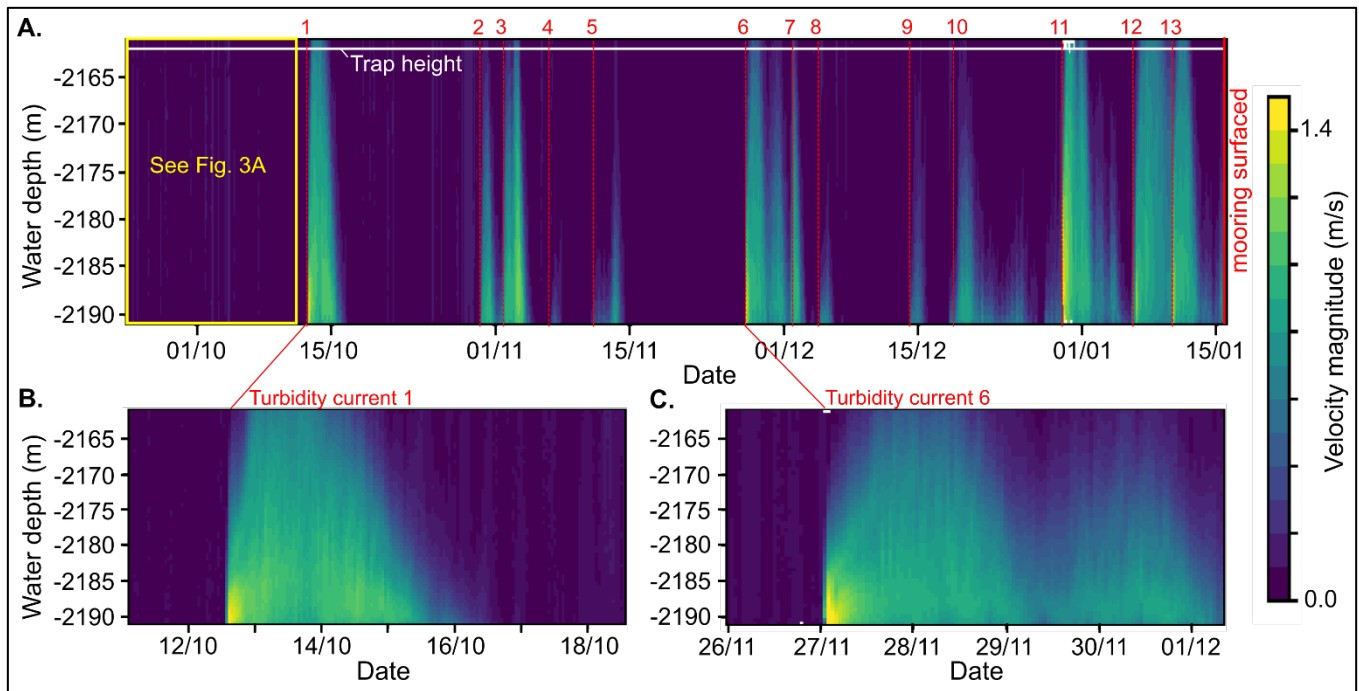

**Figure 2: Velocity magnitude measured by a 600 kHz acoustic Doppler current profiler deployed 32 m above seabed in the Congo submarine canyon (Fig. 1B for location). A.** Time series between 21$^{st}$ September 2019 and 15$^{h}$ January 2020. Numbers in red denote turbidity current events. **B.** Zoom into turbidity current event 1. **C.** Zoom into turbidity current event 6 made of two pulses.

## 3.2 Sediment trap and seabed sampling

An Anderson-type sediment trap (Anderson, 1977) was deployed on the same mooring line as our studied ADCP, at 30 m above seabed, to collect particles transported in the canyon (Fig. 1C; Maier et al., 2019). Anderson traps are made of an open top funnel of 25 cm in diameter above a 110 cm-long plastic liner tube (6 cm in diameter) allowing fallen sedimentary sequences to be preserved in a similar way to seabed sediment cores (e.g. Liu et al., 2016, Maier et al., 2019). An intervalometer unit was included in the funnel to insert discs that were scheduled to



fall every 21 days into the liner tube. The timing of trapped particles could then be precisely compared with the timing of turbidity current events detected by the ADCP (Fig. 7). Once retrieved from the sea, the trap was x-rayed

and scanned for sediment properties using a Geotek X-ray CT Core Imaging System (MSCL-XCT; Fig. 6 and Fig. A1). Seventy-eight 1-cm-thick samples were extruded from the trap liner. The top twenty-five centimetres of sediment in the trap was disturbed and thus not included in our sample set (Fig. A1).

Multi-cores (50 cm deep maximum) were collected in the canyon thalweg and on two terrace levels (at 1932 and 2007 m water depth) near the M9 mooring location in the canyon-axis (Fig. 1C). Two multi-coring sites were also

studied in the distal channel at 4795 and 4765 m water depth, located in the channel thalweg and levee respectively (Fig. 1A). These sites were used to compare the seabed carbon composition found in the upper-canyon at M9 at a water depth of 2190 m, and 800 km downstream of the M9 location along the canyon-channel axis. In total, thirteen samples from multi-cores were analysed, including three samples (45, 25 and 5 cm deep) per multi-coring site in the canyon thalweg and two terrace levels, and two samples (45 and 5 cm) per multi-coring site in the distal channel

thalweg and levee.

All ninety-one samples collected from the trap and the multi-cores were analysed for grain size using a Master-sizer laser particle size analyser, and for organic carbon geochemistry as described below.

### 3.3 Organic carbon geochemistry

### 3.3.1 Sample preparation and total organic carbon content

Samples were freeze-dried, ground into powder and rinsed with deionised water to remove salt prior to all organic carbon geochemistry measurements. To determine the total organic carbon content (TOC) of samples, between 4 and 6 mg of sample were weighed into silver capsules. Between 100 and 200 µl of 1M hydrochloric acid were added to each capsule to remove inorganic carbon (i.e. carbonates). Capsules were then inserted in a LECO elemental analyser to measure TOC and total nitrogen (TN) contents. Duplicate measurements were made for TN

and TOC analyses at the Panoply platform (Paris-Saclay, France). Triplicate TOC measurement was conducted for each sample at the FOCUS Isotopy platform, in ULiège (Belgium). The mean value between replicates is given (Table 1), and the uncertainty is taken as the standard deviation between replicates. The average standard deviation between is 0.02% for TN duplicates (n=182) and 0.30 % for TOC triplicates (n=273).



### 3.3.2 Organic carbon stable isotopes

Organic carbon stable isotopes (reported as $\delta^{13}C$ values) were measured to determine the terrestrial or marine origin of the samples (Hecky and Hesslein, 1995). Knowing the TOC of each sample, we weighed adequate masses (between 1 and 3 mg) for optimal measurement of organic carbon stable isotope by mass spectrometry. Samples were acidified with 1M hydrochloric acid in silver capsules to remove inorganic carbon. They then were measured for $\delta^{13}C$ in triplicates by isotope ratio mass spectrometry (IRMS) using a ThermoScientific Delta+XP coupled to a

Flash Elemental Analyzer 1112 Series at the Panoply Platform (Paris-Saclay, France) and an Elementar precision coupled to a VarioMicro Elementar at the FOCUS Research Unit (ULiège, Belgium). Results are reported in permil relative to the Vienna PeeDee Belemnite (VPDB) standard. Presented values correspond to the mean value between triplicates and uncertainty is taken as the standard deviation of triplicates (Table 1). The average standard deviation between $\delta^{13}C$ triplicates (n=273) is 0.37 ‰.

### 235 3.3.3 Radiocarbon

Radiocarbon ($^{14}C$) was measured on thirty samples (out of the ninety-one) to determine the $^{14}C$ composition of the organic carbon transported and deposited in the submarine canyon and channel (Table 2). The $^{14}C$ composition of organic matter relates to its age (recent, pre-aged or geologic) and is a crucial information regarding its reactivity, and thus its tendency to be mineralized if labile or be sequestered from the atmosphere for long timescales if

refractory (Hemingway et al., 2019). Radiocarbon measurements were carried out at the UK National Environmental Isotope Facility (NEIF). Sediment samples were acidified by fumigation with hydrochloric acid to remove any trace of inorganic carbon. Organic carbon was then converted to $CO_2$ through combustion and purified cryogenically following NEIF standard procedures (Ascough et al., 2024). An aliquot of the purified $CO_2$ was used to determine its $\delta^{13}C$ (‰ VPDB) by IRMS (Thermo Fisher Delta V). A further aliquot was graphitized following

NEIF procedures (Ascough et al., 2024) to measure its $^{14}C$ composition by accelerator mass spectrometry (AMS). Radiocarbon measurements were corrected for isotopic fractionation using the measured $\delta^{13}C$ value by IRMS. Radiocarbon results were reported in the form of the fraction modern ($F^{14}C$) (Reimer et al., 2004), identical to the $A_{SN}/A_{ON}$ (Stuiver and Polach, 1977) and $^{14}a_N$ (Mook and van der Plicht, 1999) metrics. Comparing the $^{14}C$ composition of samples of different calendar age is challenging because of changing atmospheric $^{14}C$ composition

with time especially for post-bomb (> 1950) samples and $^{14}C$ decay (Soulet et al., 2016, Skinner and Bard, 2022). A way to make comparable radiocarbon data obtained from samples formed at different calendar years is to normalise them to the radiocarbon composition of the atmosphere when the samples formed. This metric is called the isotopic relative enrichment $F^{14}R$ (Soulet al., 2016). Hence the radiocarbon data presented in Figs 6 and 7 are



reported as the $F^{14}R$ metric relative to the $^{14}C$ composition of the atmosphere of year 2019 (i.e., when the samples
were collected): $F^{14}R = F^{14}C_{sample}/F^{14}C_{atm\ in\ 2019}$. Radiocarbon data of samples from the literature (Spencer et al.,
2012, Hemingway et al., 2017) were similarly reported relative to the $F^{14}C$ of the atmosphere in their collection
year. The post-bomb atmospheric $^{14}C$ reference dataset is taken from Hua et al. (2022).

### 3.3.4 Rock-Eval analyses

To further characterize the types of organic carbon and compare the carbon composition of our samples with
samples from the Congo distal lobes (Baudin et al, 2017), all samples were analysed by pyrolysis and oxidation
using the Rock-Eval 6 Turbo device at the Paris Earth Science Institute (ISTeP; Baudin et al., 2015; Fig. 7 and
Table 1). The desalted samples were heated in two successive stages. The first stage corresponds to the pyrolysis
under an inert atmosphere ($N_2$), where the sample is heated at 180°C for 3 minutes and then up to 650 °C at a ramp
of 30 °C/min. The second stage is an oxidation of the residual carbon that has resisted to pyrolysis, under the
laboratory atmosphere from 300 °C to 850 °C at a temperature ramp of 20 °C/min. During the pyrolysis phase,
hydrocarbon (HC) effluents were monitored by a flame ionisation detector, and CO and $CO_2$ were monitored by
infrared detectors. During the oxidation phase, CO and $CO_2$ were monitored by infrared detectors. Hydrocarbon
detected during the pyrolysis stage are expressed in mg HC/g of sample, whereas $CO_2$ evolved during the pyrolysis
stage up to 400°C are expressed in mg $CO_2$/g sample. Dividing the amount of HC and $CO_2$ by the total organic
carbon content (TOC) of each sample provides the hydrogen index (HI in mg HC/g TOC) and the oxygen index
(OI in mg $CO_2$/g TOC), respectively. HI and OI can inform on the type of organic matter (e.g. terrestrial, oxidised,
marine) in a HI versus OI diagram (Baudin et al., 2015, 2017).

## 4 Results

### 4.1 Observations of particle transport in the Congo submarine canyon

Downslope turbidity currents in the upper canyon occurred for 30% of the monitoring period duration (between
21st December 2019 and 15th January 2020), which is consistent with previous studies in the area (Azpiroz-Zabala
et al., 2017, Simmons et al., 2020). Eight flows were detected at the mooring M9 between 21st September and 10th
December 2019 (Fig. 2) and terminated in the upper canyon at < 200 km off the Congo River mouth (Talling et al.,
2022). These events have internal velocity magnitudes of maximum 1.5 m/s (Fig. 2), and frontal velocities of up to
4 m/s based on transit times between moorings (Talling et al., 2022). Flow durations varied between < 1 day (e.g.,
Flow 4 in Fig. 2) and > 3 days (e.g., Flows 1 and 6 in Fig. 2A, C). Flow heights reached > 32 m (i.e., height of the
ADCP instrument above canyon floor) in four of the eight observed turbidity currents. We note that our eight





studied flows have shorter run-out distances and slower speeds compared to the January 2020 event which ran out for >1000 km down canyon (Talling et al., 2022). This latter flow broke our M9 mooring line so it could neither

be captured in the ADCP nor in the sediment trap.

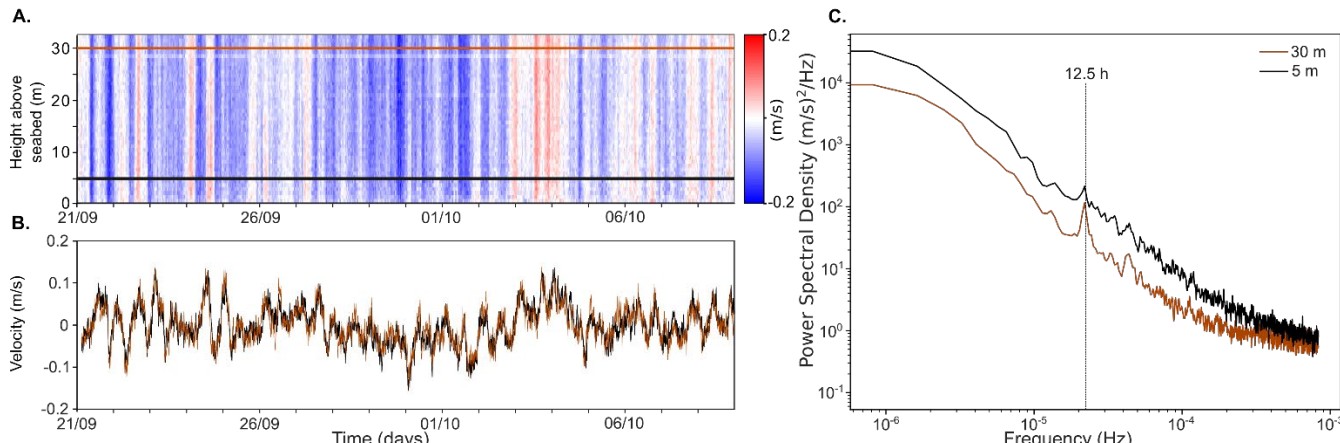

**Figure 3: Velocity along canyon (V$_{along}$) and frequency analysis at mooring M9 at 2,170 m water depth, for the period 21$^{st}$ September to 10$^{th}$ October 2019 when there was no turbidity current activity (see Fig. 2A for context). A.** Time
series of V$_{along}$ between 21$^{st}$ September and 10$^{th}$ October 2019. The yellow and black horizontal lines indicate the heights above seabed of velocity time-series displayed in B. **B.** V$_{along}$ speeds at 5 and 30 m above canyon floor. **C.** Power Spectral Density computed for the whole time serie (21/09/2019 to 15/01/2020) at 5 m and 30 m above the seafloor.

Along canyon velocity (V$_{along}$; Figs. 3, A2, and A3) computed before and between turbidity current events, reveals that there is an oscillatory continuous flow regime with currents alternating between net-upslope (negative values)
and downslope (positive values) directions, and with speeds ranging from -0.15 to +0.15 m/s. For this background 'oscillatory flow', the residual V$_{along}$ current direction points upslope as illustrated by the hodographs computed for the arrays 5 and 30 m above seabed (Fig. 4). Across-canyon velocity (V$_{across}$) oscillates between -0.15 (SW direction) and +0.15 (NE direction) m/s but has little influence on the net current trajectory compared to V$_{along}$ and shows no pattern or dominant frequencies. This oscillatory flow is homogenous through the observed water column
(i.e., 30m), as demonstrated by constant values in both V$_{along}$ and V$_{across}$ through the ADCP measured range (32 m; Figs. 3A, A2 and A3). The CTD data collected when there was no turbidity current activity also reveals that salinity, oxygen and turbidity values are homogenous, and relatively high in the confined water column, compared to intermediate unconfined waters (Fig. 5). The frequency analysis reveals one main frequency of 12.5 h (Fig. 3), corresponding to semi-diurnal tides in the Congo River estuary. The tidal signal is composed of external tides and
internal tides. External tides are forced by the gravitational interaction of the Sun and the Moon with Earth's oceans; they feature currents that are constant throughout the water column. Internal tides are internal gravity waves at tidal frequency, generated by the interaction of the tidal currents with the seafloor topography (e.g., Garrett and Kunze,





2007); they feature sheared vertical structures of currents and pressure. With the ADCP measurements alone, we cannot unambiguously attribute the semidiurnal signal to internal tides, although they likely dominate the signal.

We thus call the observed oscillations confined within the canyon morphology 'tides' in the following text.

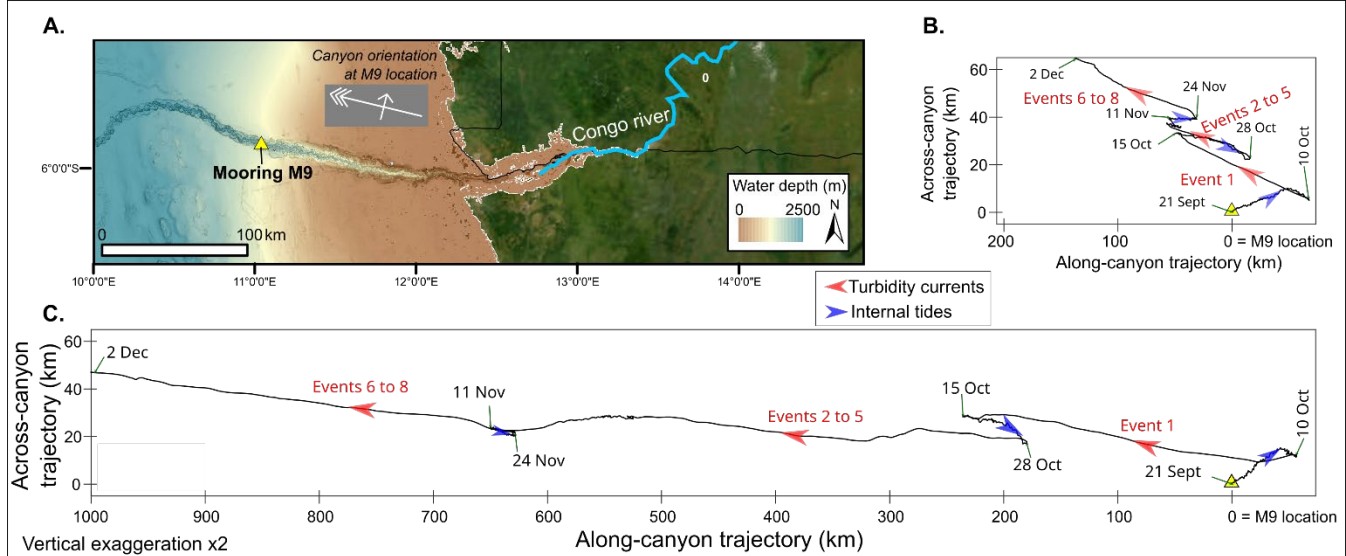

**Figure 4. A.** Location of the moored ADCP (Mooring M9) deployed in the Congo Canyon at 2190 m of water depth. Aerial image of the Congo River mouth is taken from ©Google Earth. **B.** Hodograph showing residual currents measured at mooring M9 location, 30 meters above seabed. **C.** Hodograph showing residual currents measured at mooring M9 location, 5 meters

above seabed. The start and end dates of turbidity current series recorded by the ADCP are marked. Red arrows show a downslope trajectory due to turbidity currents, whereas blue arrows show a net upslope trajectory due to internal tides.

Both tides and turbidity currents measured by the ADCP are recorded in the sediment trap deployed just below the ADCP instrument at 30 m above canyon floor. The trap contains a 1.2 m thick sedimentary succession made of

mud (mean $D_{90}$ = 22 μm) and coarse silt to very-fine sand layers (mean $D_{90}$ = 51 μm; Fig. 6). These silt to very-fine sand layers are between 2 and 7 cm thick. The disks released by the intervalometer every 21 days, allow us to correlate the base of the fine-sand layers with the turbidity current events detected by the ADCP (Figs. 6A, B). The trap was deployed 30 m above the seabed, hence particles transported closer to the seabed could not be retrieved. Despite this, the base of the silt to very-fine sand event layers observed in the trap are sharp and overlain by a

fining-upward succession (Fig. 6C), features typical for the deposits of turbidity currents (i.e., Bouma sequence; Bouma, 1962). Turbidity currents that were detected close together in the ADCP data (e.g. events 2 to 5 and events 6 to 8) appear as multi-pulses in the grain size analysis measured on the trapped sediments (Fig. 6C).



Finally, sedimentary facies observed in the five multi-cores from the canyon thalweg, terrace levels and channel
correspond to homogenous muds, without any structure such as laminations. Samples collected in these multi-cores
have a mean $D_{90}$ of 28 µm, and are thus of similar grain size to the muds observed in the trap (Fig. 7A,B).

**Figure 5.** Conductivity Temperature Depth (CTD) profiler and turbidity sensor data collected on 21$^{st}$ September 2019 at
mooring M9 location, at 2170 m of water depth (see Fig. 1 for location) outside of turbidity current activity.





**Figure 6. Overview of the data collected from the sediment trap. A.** Velocity magnitude measured by the ADCP averaged over 30 m above seabed. Flows recorded in ADCP data (Fig. 2) are numbered. **B.** X-ray of the sediment succession retrieved from the sediment trap deployed on the same mooring as the ADCP, 30 m above canyon floor. **C.** Grain size measurements made on samples taken every 1cm in the trap succession. **D.** Total organic carbon content (TOC) and organic carbon over nitrogen ratios (C/N) measured on samples taken every centimetre along the trapped succession. **E.** Carbon stable isotope ratio ($\delta^{13}C$) and relative $^{14}C$ enrichment (relative to year 2019; $F^{14}R$, Soulet et al., 2016) measured on samples taken every centimetre along the trapped succession.

### 4.2 Organic carbon composition in trapped and seabed sediments

Sediment in the trap shows a mean TOC content of 4.0 ± 0.6 % (n = 78), with a minimum and maximum TOC of 1.8 and 4.6 % respectively (Fig. 6D). TOC content in the trap appears to be controlled by sediment grain size, with sandy samples ($D_{90} > 63$ μm) having lower TOC (mean = 2.6 ± 0.4 %, n=9) compared to muddy samples (mean =



4.2 ± 0.4 %, n=69). TOC found at the base of the very-fine-sand event layers is thus relatively low compared to the

top of each event layer (Fig. 6D). Homogenous muds below sandy event beds show a slight decrease in TOC content with time at equal D50 and slightly decreasing D90 (e.g., from 36 to 30 cm, TOC = 4.5 ± 0.3 to 3.8 ± 0.3 % and from 83 to 75 cm, TOC = 4.4 ± 0.3 to 3.8 ± 0.2 % in Fig. 6D). TOC measured on the multi-core muddy samples from the surficial canyon seabed reveal a slightly lower organic carbon content (TOC = 3.6 ± 0.4 %, n=11 Fig. 6A) compared to the muddy sediment observed in the trap. TOC values from the distal channel seabed (TOC

= 4.3 ± 0.7 %, n=4) are similar to those in the trap. Overall, there is a negative relationship between TOC and grain size in both the trapped and seabed samples (Fig. 7A).

All trap and multi-core samples analysed here contain organic carbon of terrestrial origin based on rather constant low carbon stable isotope ratios ($\delta^{13}$C = -26.8 ± 0.3 ‰, n = 91), high C/N ratios (17.1 ± 1.4, n = 87), and low hydrogen and oxygen indexes (HI = 161 ± 9 mg HC/g TOC, and OI = 220 ± 34 mg $CO_2$/g TOC, n = 91).

Radiocarbon measurements ($F^{14}R$ values) conducted on 30 samples from both the trap and multi-cores (in both canyon and channel) range between 0.791 and 0.949. There is no significant relationship between grain size and radiocarbon isotope composition based on our 30 measurements, although it appears that the four oldest radiocarbon ages ($F^{14}R$ = 0.791 to 0.829) measured in our sample set correspond to trap sandy and distal channel samples (Fig. 7B).







**Figure 7. Measurements made on trap and seabed samples collected in the Congo Canyon and Channel, with previous data from Congo River and lobe. A.** Total organic carbon content (TOC) against grain size $D_{90}$ in micrometres. **B.** Relative $^{14}$C enrichment (relative to year of sample collection in 2019; $F^{14}R$) against carbon stable isotope ratios ($\delta^{13}C$). River data are taken from Spencer et al. (2012) and Hemingway et al. (2017) and are averaged between samples collected in one year (labelled 2008, 2011, 2012 and 2013). Hydrogen index against oxygen index derived from Rock-Eval pyrolysis/oxidation. Lobe data are taken from Baudin et al. (2017).



## 4.3. Organic carbon flux calculations

Combining timing and durations of turbidity currents from the ADCP data with sediment properties and carbon
content measured in the trap, we calculated transit organic carbon fluxes for our study period as follows (Fig. 8).
First we derived sedimentation rates (i.e., rates at which particles were falling in the trap) for periods when turbidity
currents were active and for periods when tides were dominant. This could be achieved by matching the date/time
of four scheduled disks fallen in the sediment trap with the corresponding date/time of flows recorded by the ADCP.
To refine rates related to periods of active turbidity currents, the base of very-fine sand layers identified in the trap
(e.g., at 99, 70 and 39 cm of depth in Fig. 6) were correlated to the start of turbidity currents 1, 2 and 6. Once
sedimentation rates were known for each period, we used equation (4) to derive a specific organic carbon flux per
sample, expressed in grams of organic carbon per squared meter and per day (g $C_{org}$/m$^2$/day).

$$\text{Flux } C_{org} = \frac{\text{TOC x } (1 - \varphi) \text{x } \rho_{quartz} \text{ x SR}}{SA_{funnel}\big/SA_{liner}} \qquad (4)$$

Where TOC is the total organic carbon content measured by elemental analysis (see section 3.2), SR is the
sedimentation rate obtained as explained above, $\varphi$ is the porosity of the sediment fallen in the trap measured by the
Multi-Sensor-Core-Logger (Fig. S1) and $\rho_{quartz}$ is the density of quartz (2.65 g/cm$^3$). $SA_{liner}$ is the surface area of the
core liner (diameter of 6.3 cm) containing the trapped particles, and $SA_{funnel}$ is the surface are of the open top funnel
(diameter of 25 cm).

Using this approach, we find that transit fluxes of organic carbon at 30 m above seafloor in the Congo Submarine
Canyon range between 5 and 33 g $C_{org}$/m$^2$/day. Fluxes attributed to turbidity currents are thus on average two to
three times higher compared to the carbon fluxes related to tides (Fig. 8). We note that these are sediment fluxes at
30 m above the bed, and may thus differ from overall sediment fluxes down the canyon, especially due to vertical
variations in sediment concentrations and velocities expected in turbidity currents as discussed in section 5.1.





**Figure 8. A.** Velocity magnitude measured by the ADCP averaged over 30 m above seabed. **B.** Apparent sediment accumulation rates (in cm/day) calculated in the sediment trap, and X-ray of the trapped core. **C.** Transit fluxes of particulate organic carbon (in g $C_{org}/m^2$/day) calculated based on B., total organic carbon contents (Fig. 6C) and sediment properties measured in the trapped sediment by multi-sensor-core-logging.



## 5 Discussion

### 5.1 What are the hydrodynamic processes controlling the transport of particulate organic carbon in the Congo submarine canyon?

Two processes drive the transport of sediment and particulate organic carbon through the Congo Submarine Canyon from the Congo River estuary to the deep-sea fan (Fig. 9): downslope turbidity currents and background tides.

Turbidity currents occur for 35% of the time during monitoring periods in the upper Congo Submarine Canyon (Cooper et al., 2013, Azpiroz-Zabala et al., 2017, Simmons et al., 2020; Talling et al., 2022). Through our study, we show that turbidity currents carry organic carbon at a much higher transit flux (two to three times higher, Fig. 8) compared to the transit flux associated with tides, at least for relatively small turbidity currents (see Azpiroz-Zabala et al., 2017; Talling et al., 2022) and at a height of 30 m above the bed (Fig. 2A). Our calculated fluxes should not be regarded as sedimentation rates as they represent a settling flux in the sediment trap, which greatly exceeds the settling rate on the seabed (Gardner, 1980), particularly in highly active submarine canyons. Despite this, the transit flux due to turbidity currents is likely underestimated because the height of our sediment trap (30 m above seabed) did not allow sands transported at the base of turbidity currents to be collected. Muddy sand event beds associated with large amount of vegetation debris (TOC of up to 11.3 %) were reported in some piston cores collected in the Congo Canyon and Channel, together with clean sandy beds devoid of any vegetation debris (Baudin et al., 2010, Baker et al., in review). The base of turbidity currents (e.g., 0 to 5 m above seabed), usually characterized by higher velocity and sediment concentration (Kneller and Buckee, 2000, Sequeiros et al., 2009), can carry fresh vegetation debris associated with sands. Vegetation-rich sandy turbidites were observed in submarine canyons deposits elsewhere (e.g., Saller et al., 2006, Lee et al., 2019, Hage et al., 2020), supporting our hypothesis. Furthermore, the hodograph of velocity data presented here from 5m above the bed (Fig. 4) suggest that turbidity currents in 2019 carried > 10 times the sediment flux in tides, assuming that tides and turbidity currents have the same sediment concentrations. As the sediment concentration within the turbidity currents was likely greater than that in tidal flows, as turbidity currents are driven by the sediment they suspend, this may also be a significant underestimate. Turbidity currents are also highly episodic, and far most sediment may be carried in much larger and infrequent canyon flushing flows, as occurred in January 2020, breaking our mooring line (Fig. 2). Just two canyon flushing turbidity currents in 2020 likely carried ~2.65 km3 of sediment down the Congo Canyon, which is equivalent to 19-33% of the total annual sediment flux from all of the world's rivers (Talling et al., 2022). Those much larger canyon flushing flows may have recurrence intervals of 20-50 years (Talling et al.,



2022). In contrast to heterogeneous TOC contents observed within sandy deposits from the base of turbidity currents, our study shows that the upper part (> 30 m) of turbidity currents carries muddy sediment with homogenous, high TOC content (mean = 4.2 ± 0.4 %, n=69) and silt to very-fine sands with lower TOC content (mean = 2.6 ± 0.4 %, n=9) compared to the muds.


Between episodic turbidity currents, we observe a background oscillatory flow with dominant frequencies related to semi-diurnal tides (12.49 h of period), at a speed of up to 0.15 m/s, directed both up and down canyon. These tidal currents affect the measured (32 m) water column homogenously and show a residual current direction that points upslope (Figs. 3, 5 and 6). Tidal currents and their residual upslope flux were described previously in the

Congo distal submarine channel at ~4000 m water depth based on current meter data, but with slightly lower speeds compared to our measurements (-0.1 to +0.1 m/s; Vangriesheim et al., 2009). The V-shaped morphology of submarine canyons and channels appears to enhance the vertical mixing up and down canyon compared to the higher open water column. Currents associated to tides are thus morphologically confined closer to the canyon thalweg (i.e. tides are *internal* to the canyons), as observed for example in the Logan (Eastern Canada) and

Cassidaigne (Mediterranean Sea) submarine canyons (Li et al., 2019, Brun et al., 2023). Confined mixing by tides keeps in suspension the fine particulate organic carbon delivered via the slower-moving, muddy tail of turbidity currents. This is attested by the presence of muds in the sediment trap between the turbidity current event beds, falling at rates of 0.7 to 1.1 cm/day during periods dominated by tides (Figs. 6 and 8B). Seabed deposits of tides have been described in the literature as being composed of laminated muddy sand to silt (e.g., Normandeau et al.,

2023). Such laminations cannot be produced in a sediment trap and are thus not observed in our trapped succession (Fig. 6). The upper 30 cm of the canyon and channel seabed were retrieved in the multi-cores, which do not reveal laminations either (at least to the naked eye). The flushing and reworking of sediment by turbidity currents likely prevent any fine-scale laminated structures from being preserved on the seabed, confirming again that turbidity currents are the dominant hydrodynamic process in the Congo Submarine Canyon.


In summary, turbidity currents rapidly move sediment and particulate organic carbon downslope in one go at a transit flux that we estimate to be ten times higher compared to the flux induced by tides. Moreover, once infrequent and much more powerful canyon flushing turbidity currents are included in sediment fluxes, turbidity currents likely transfer even more sediment than tides. It then appears that the fine material flushed episodically by the top

part of turbidity currents into the canyon is then kept in suspension and mixed in the canyon water column by tides.





The impact of these combined hydrodynamic processes on organic carbon composition (i.e., isotopic composition and Rock-Eval indexes) is further discussed in the next section.

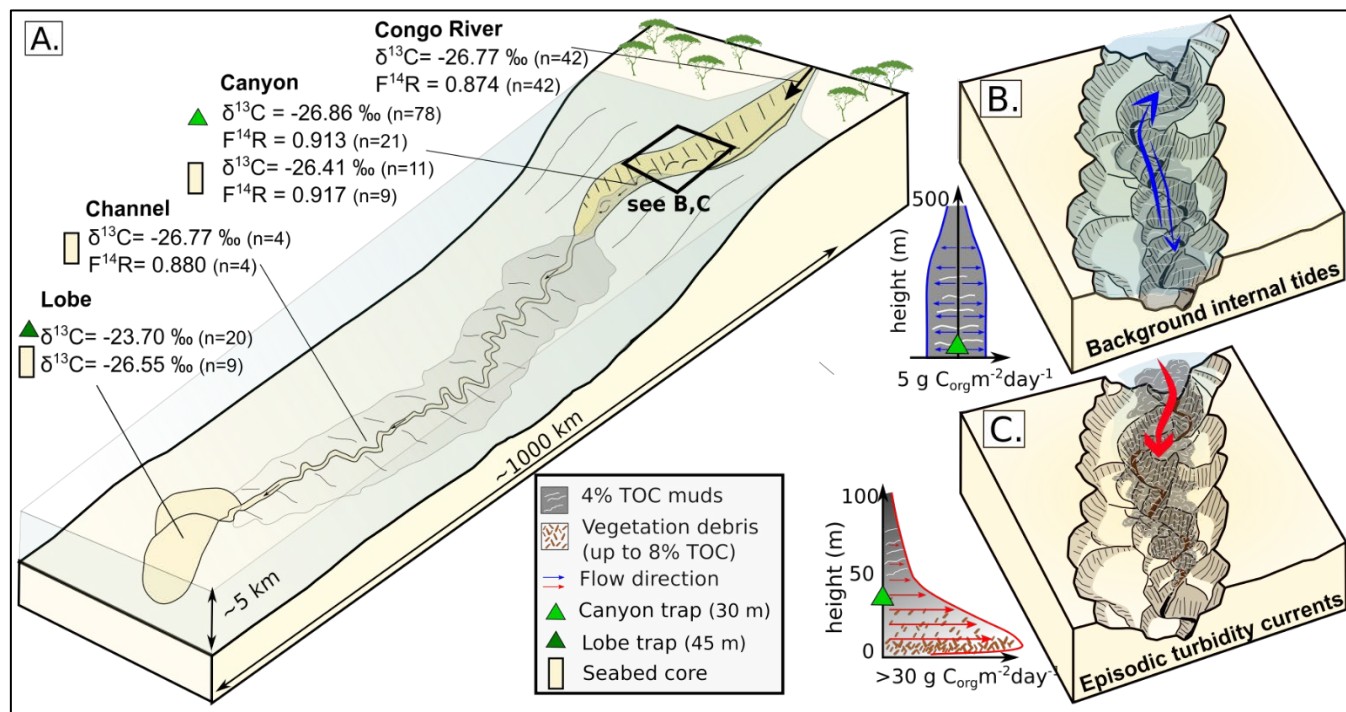

**Figure 9. Summary of particulate organic carbon transport in the Congo Submarine Fan. A.** Schematic of the Congo
River to submarine canyon to deep-sea fan system (modified from Babonneau, 2002). Carbon stable isotopes and radiocarbon compositions are shown for each part of the fan, with data river data taken from Spencer et al. (2012), Hemingway et al., 2017, and lobe date taken from Baudin et al. (2017). **B.** 3D illustration of internal tides observed in the canyon with a net residual flow that is directed up-canyon, and idealized velocity profile showing the oscillations with a downslope (positive values) and upslope direction (negative values). Particulate organic carbon-rich muds are also represented. **C.** 3D illustration of a turbidity
current transporting particulate organic carbon down canyon, and idealized velocity profile of a turbidity current showing coarse woody debris at the base of the flow and carbon-rich muds within the top of the flow. Carbon flux (in g $C_{org}/m^2/day$) correspond to the estimated transit flux based on the sediment trap at 30 m above seabed.

**5.2 How does carbon transiting the canyon compare with carbon composition in the river and deep-sea fan?**

The trap and seabed sediments in our study show similar carbon isotopic (both $\delta^{13}C$ and $^{14}C$) compositions to those
previously sampled in the Congo River (Figs. 7B, 9; Spencer et al., 2012, Hemingway et al., 2017), implying that the material transiting in the submarine canyon and channel is entirely river-derived. Total organic carbon contents measured in the Congo River suspended sediment averages 6.1 ± 1.0 % with maximum TOC of 8.9 % (Mariotti et al, 1991, Coynel et al., 2005, Hemingway et al., 2016). This is higher than the average TOC measured in our canyon trap (TOC = 4.0 ± 0.6%), yet the Congo River bedload was not considered in those previous studies and could





likely decrease the river average TOC. Furthermore, TOC of up to 11.3 % was measured in vegetation-rich muddy
       sand deposits in piston sediment cores retrieved from the Congo Submarine Canyon axis (Baker et al., in review).
       Such vegetation-rich deposits are not present in our canyon trap that is located 30 meters above seabed.

       Carbon stable isotopes and Rock-Eval index compositions (HI and OI) are also similar between our submarine
       canyon/channel samples, and previously reported seabed samples retrieved in distal lobes (Fig. 7D, Fig. 9; Baudin
et al., 2010, 2017, Stetten et al., 2015). This suggests that organic carbon is efficiently flushed to deep-sea lobes,
       yet this hypothesis needs to be further corroborated by radiocarbon measurements on particulate organic carbon
       deposited in lobes. Our canyon and channel samples show a different carbon composition compared to sediment
       retrieved from a trap previously deployed 45 m above the seabed at 4779 m of water depth (Rabouille et al., 2017a,
       Baudin et al., 2017). This trap shows higher $\delta^{13}C$, higher HI and lower OI, compared to our canyon and channel
samples (Figs. 7D and 9). The organic carbon composition of the distal trap corresponds to marine organic carbon
       sinking from the photic zone in the water column at 45 m above seabed. This marine signature was not found in
       the seabed sediments beneath the same trap, probably due to rapid degradation of marine organic matter in the first
       few centimetres below the seabed (Treignier et al., 2006, Baudin et al., 2017). There is no evidence for the presence
       of such marine organic carbon in our canyon trap and seabed samples, probably because terrestrial organic carbon
sourced from the Congo River brought in by turbidity currents is dominant and dilutes the marine signal in the
       canyon. Turbidity currents were not recorded in the distal sediment trap (Rabouille et al., 2017a, Baudin et al.,
       2017) as flows reach the distal channel to lobe transition zone at a much lower frequency (every 6 to 10 years;
       Dennielou et al., 2017), compared to the upper canyon where flows occur several times per year (e.g., Fig. 2A).

       Overall, the isotopic composition and total content of particulate organic carbon in the Congo Canyon during our
studied period is similar to both that reported in the Congo River (Spencer et al., 2012, Hemingway et al., 2017),
       and in the deep-sea lobe seabed (Baudin et al., 2010, 2017, Stetten et al., 2015). Both tides and episodic turbidity
       currents thus do not seem to significantly affect the composition of riverine particulate organic carbon transiting
       through the submarine canyon towards the deep-sea (Fig. 9). Further, tides appear to mix fine particulate organic
       carbon, leading to relatively homogenous stable carbon and radiocarbon isotope composition in the trap (Fig. 6).
Despite this overall homogenous trend, we note that carbon contents in the tidal muds show a slight decreasing
       trend (TOC = 3.8 ± 0.3 %, n = 12) below coarser deposits and compared to muds associated with turbidity current
       events (TOC= 4.2 ± 0.4 %, n=69; Fig 6D) in the trap. This occurs at relatively constant grain size D50 ($D_{50} = 5.9$
       ± 0.6 µm, n = 12) but slightly decreasing D90 (e.g., from 24 to 18 µm between 37 and 30 cm in the trap; Fig. 6D).
       This subtle decrease in TOC in the tidal muds can be explained by two reasons. First it could represent loss of




organic matter (up to ~10% total loss in the tidal muds), which could occur during suspension in the canyon where the water column is oxygenated as shown in the CTD data (up to ~500 m above seabed; Fig. 5). However, high rates of organic matter degradation are known for fresh marine organic matter in the laboratory, rather than terrestrial organic matter (Middelburg, 1989, Prahl et al., 1997). Second, the decrease in TOC in tidal muds could also be explained by hydrodynamic sorting whereby free organic matter (i.e., non-mineral organic carbon) settles

in the water column together with silty to sandy particles. This second hypothesis is supported by a slight decrease in the grain size D90 curve in the tidal muds (Fig. 6C, D), implying there is no loss of total organic carbon content at a given grain size.

**5.3 How does carbon transport in the Congo Submarine Canyon compare with other submarine canyons?**

It is challenging to monitor directly sediment transport in submarine canyons due to difficulty of access and to

potential damage of measuring instruments by powerful flows (e.g., Khripounoff et al., 2003, Sumner et al., 2012). However, advances in tools used to measure sediment transport have enabled detailed direct observations in a few canyons globally (Liu et al., 2016, Paull et al., 2018, Clare et al., 2020, Talling et al., 2023). Hydrodynamic processes such as turbidity currents and tides have been reported in canyons elsewhere, but sometimes with different flow properties compared to those in the Congo Canyon. Turbidity current frontal speeds can vary between

< 0.5 m/s (e.g., Cassidaigne canyon, Brun et al., 2023) to > 19 m/s (e.g., Grand Banks, Heezen and Ewing, 1952). Flow durations of several days in the Congo Canyon are by far the longest flow durations yet observed, with other flows lasting for a few minutes to hours (e.g., Paull et al., 2018; Heijnen et al., 2022). Turbidity currents in the upper Congo Canyon also occur for ~30% of the time during monitoring experiments, which is a far higher fraction than in other monitored systems (Talling et al., 2023). Oscillatory flows due to tides are relatively slow in the

Congo Canyon (<0.15 m/s). In other settings, the maximum speed of tides can reach 1 m/s (e.g., Nazaré Canyon, Masson et al., 2010; Bay of Biscay Canyons, Mulder et al., 2012, Guiastrennec-Faugas, 2020; Gaoping Canyon, Liu et al., 2016; Monterey Canyon, Maier et al., 2019; Cassidaigne Canyon, Brun et al., 2023; Whittard Canyon, Heijnen et al., 2022).

In contrast to the growing number of studies reporting observations of sediment transport in submarine canyons,

the impact of hydrodynamic processes on carbon composition in submarine canyons is poorly constrained. A few studies compared particulate organic carbon composition in traps and seabed sediments in submarine canyons (e.g., Kao et al., 2014, Liu et al., 2016, Maier et al., 2019). In a similar way to the Congo Canyon, trap and seabed sediments in the Gaoping Canyon, offshore Taiwan, include particulate organic carbon dominated by inputs from





the Gaoping River whose mouth is very close to the canyon head (Kao et al., 2014). The efficient transport of this
terrestrial organic carbon is also controlled by episodic downslope turbidity currents, as well as tides with a net up-
canyon direction but that are much faster (1 m/s) compared to Congo (Liu et al., 2016). These processes transport
a mix of coarse, plant-derived organic matter, soil organic matter, and a large component of rock organic carbon
(Hilton et al., 2012). Unlike the Congo and Gaoping Canyons, Monterey Canyon is not directly fed by a river, but
its head is located next to the shoreline. Particulate organic carbon composition in the Monterey Canyon trap-
samples is associated with fast (up to 1 m/s) tidal currents, and comprise a mixture of marine and terrestrial carbon
(Maier et al., 2019). This trap composition contrasts with seabed samples collected in Monterey Canyon that are
dominated by terrestrial carbon brought by turbidity currents (Maier et al., 2019). Thus, the marine organic carbon
seen in the Monterey trap samples is not preserved on the seabed over long time scales, as seen for distal lobe areas
in the Congo Fan by Baudin et al. (2017).

Particulate organic carbon concentrations in Congo Canyon sediments are very high (4.0 ± 0.6%), compared with
trapped sediment in the Monterey Canyon (TOC = 1.9 ± 0.3%; Maier et al., 2019) and with the Gaoping Canyon
sediments (TOC = 0.5 ± 0.3 %; Kao et al., 2014). In the case of the Gaoping Canyon, the difference relates to the
starkly different geomorphic setting of the river input basins. In Taiwan, short and steep, small mountain
catchments have rapid erosion rate, producing high rates of terrestrial organic carbon transfer, but with a low overall
TOC due to the deep erosion depth and clastic sediment input (Hilton et al., 2012). In contrast, the Congo Basin
drains vast lowland regions before entering the ocean, with high terrestrial productivity and the presence of tropical
peatlands (Coynel et al., 2005). There, erosion processes do not typically mobilise unweathered rocks, and thicker
soils can be sources for the particulate organic carbon load (Hemingway et al., 2017). Overall the rates of particulate
organic carbon transfer per unit catchment area are lower in the Congo River than in Taiwanese rivers (Galy et al.,
2015), but the %TOC of the river sediments and the total fluxes are much higher. Monterey Canyon represents an
intermediate end member. There, small catchments deliver sediment and particulate organic carbon to the coastal
zone, but the canyon itself is not tightly connected to any river. Higher %TOC in that setting reflect a more
important role for marine organic matter (Paull et al., 2006).

More generally, TOC in the Congo Submarine Canyon is much higher than the global average of continental shelf
sediments (mean TOC = 1.63 %; van der Voort et al., 2021). Erosion of organic carbon-rich material in the Congo
watershed and subsequent transport by the Congo Submarine Canyon and Channel appears to occur efficiently and
quickly due to the direct connection between the Congo River estuary and the deep-sea canyon head. Overall, our
observations of particulate organic carbon transport in the Congo Submarine Canyon reveal that strong



hydrodynamic processes in canyons appear to remove efficiently particulate organic carbon from the terrestrial
biosphere to the deep-sea. We suspect that similar types of hydrodynamic processes in submarine canyons
elsewhere, either disconnected from land (e.g., Whittard Canyon, Heijnen et al., 2002) or close to large river
systems (e.g. Swatch of No Ground Canyon close to the Ganges-Brahmaputra Rivers, French et al., 2018) may also
promote the efficient transport of terrestrial organic carbon beyond continental shelves (Talling et al., 2024).
Quantification of particulate organic carbon fluxes to the deep-sea may thus increase global estimates of ocean
carbon burial that are to date mainly focused on coastal areas (Reigner et al., 2021).

## 6    Conclusion

In this study, we show that the transport of particulate organic carbon in the Congo Submarine Canyon is controlled
by two hydrodynamic processes. First, fast-moving (up to 8 m/s) turbidity currents transport river-derived
particulate organic carbon to the deep-sea for 35 % of the time during our monitoring period (4 months). Second,
tidal currents within the confined morphology of the canyon induce a background oscillatory flow at speeds of
maximum 0.15 m/s with an up-canyon residual current direction. Tides constantly mix and oxygenate the canyon
water column, keeping fine particulate organic carbon in suspension with transport rates of 5 g C/m$^2$/yr. Downslope
turbidity currents carry both muddy and sandy particulate organic carbon at rates of 15-30 g C/m$^2$/yr, which thus
greatly exceeds the sediment transit flux by tides at 30 m above the canyon floor. We infer this flux if even ten
times higher within the lower parts of turbidity currents due to density being higher closer to the seabed (Kneller
and Buckee, 2000). Carbon isotopic composition appears consistent between the Congo River (Spencer et al., 2013,
Hemingway et al., 2017), the submarine canyon and channel, and the distal deep-sea lobes located 1200 km from
the river mouth at 5 km of water depth (Stetten et al., 2015, Baudin et al., 2017). Such small changes in age and
type of organic carbon imply that transfer of terrestrial organic carbon is efficient from river to upper canyon, and
then to the deep-sea fan. This leads to some of the highest terrestrial organic carbon contents ever observed in shelf
sediments and deep-sea fans globally.



**Table 1**. Grain size and organic carbon geochemistry measurement made on all samples collected in this study.

| Sample number | Sample code | D₅₀ (µm) | D₉₀ (µm) | TN (%) | | TOC (%) | | δ¹³C (‰) | | HI mg CO₂/g C$_{org}$ | OI mg HC/g C$_{org}$ |
|---|---|---|---|---|---|---|---|---|---|---|---|
| | | | | mean | std | mean | std | mean | std | | |
| 1 | JC187-ST-22-23 | 6 | 22 | 0,30 | 0,03 | 4,4 | 0,4 | -26,9 | 0,4 | 168 | 314 |
| 2 | JC187-ST-23-24 | 6 | 23 | 0,29 | 0,00 | 4,5 | 0,5 | -26,9 | 0,5 | 158 | 233 |
| 3 | JC187-ST-24-25 | 7 | 29 | 0,30 | 0,02 | 4,3 | 0,3 | -26,9 | 0,4 | 161 | 221 |
| 4 | JC187-ST-25-26 | 7 | 27 | 0,29 | 0,01 | 4,5 | 0,2 | -26,9 | 0,4 | 157 | 221 |
| 5 | JC187-ST-26-27 | 6 | 26 | 0,29 | 0,00 | 4,5 | 0,3 | -26,9 | 0,4 | 157 | 214 |
| 6 | JC187-ST-27-28 | 9 | 40 | 0,29 | 0,00 | 4,3 | 0,3 | -26,9 | 0,3 | 164 | 205 |
| 7 | JC187-ST-28-29 | 8 | 38 | 0,29 | 0,02 | 4,4 | 0,3 | -26,8 | 0,4 | 163 | 232 |
| 8 | JC187-ST-29-30 | 12 | 53 | 0,27 | 0,01 | 4,1 | 0,3 | -26,8 | 0,3 | 172 | 194 |
| 9 | JC187-ST-30-31 | 18 | 56 | 0,23 | 0,01 | 3,4 | 0,3 | -26,7 | 0,4 | 164 | 209 |
| 10 | JC187-ST-31-32 | 15 | 57 | 0,29 | 0,03 | 4,2 | 0,2 | -26,9 | 0,3 | 141 | 202 |
| 11 | JC187-ST-32-33 | 5 | 22 | 0,27 | 0,01 | 4,1 | 0,4 | -26,8 | 0,4 | 157 | 242 |
| 12 | JC187-ST-33-34 | 5 | 24 | 0,26 | 0,00 | 4,0 | 0,3 | -27,0 | 0,3 | 150 | 242 |
| 13 | JC187-ST-34-35 | 11 | 47 | 0,24 | 0,02 | 3,9 | 0,3 | -26,9 | 0,4 | 166 | 216 |
| 14 | JC187-ST-35-36 | 9 | 47 | 0,27 | 0,01 | 4,2 | 0,2 | -26,9 | 0,3 | 163 | 217 |
| 15 | JC187-ST-36-37 | 7 | 33 | 0,27 | 0,04 | 4,3 | 0,7 | -26,9 | 0,3 | 166 | 214 |
| 16 | JC187-ST-37-38 | 7 | 39 | 0,25 | 0,00 | 4,2 | 0,3 | -26,8 | 0,4 | 157 | 254 |
| 17 | JC187-ST-38-39 | 22 | 51 | 0,17 | 0,00 | 2,7 | 0,3 | -26,7 | 0,5 | 153 | 247 |
| 18 | JC187-ST-39-40 | 34 | 74 | 0,14 | 0,01 | 2,4 | 0,2 | -26,5 | 0,7 | 157 | 212 |
| 19 | JC187-ST-40-41 | 31 | 71 | 0,13 | 0,00 | 2,2 | 0,2 | -26,7 | 0,4 | 158 | 204 |
| 20 | JC187-ST-41-42 | 33 | 73 | 0,14 | 0,00 | 2,5 | 0,4 | -26,6 | 0,5 | 148 | 224 |
| 21 | JC187-ST-42-43 | 31 | 79 | 0,17 | 0,02 | 2,9 | 0,1 | -26,6 | 0,5 | 165 | 193 |
| 22 | JC187-ST-43-44 | 38 | 89 | 0,16 | 0,01 | 2,8 | 0,1 | -26,7 | 0,4 | 154 | 187 |
| 23 | JC187-ST-44-45 | 38 | 84 | 0,18 | 0,03 | 3,3 | 0,2 | -26,6 | 0,5 | 162 | 187 |
| 24 | JC187-ST-45-46 | 5 | 18 | 0,26 | 0,01 | 3,8 | 0,3 | -26,6 | 0,3 | 168 | 226 |
| 25 | JC187-ST-46-47 | 5 | 14 | 0,28 | 0,00 | 3,9 | 0,2 | -26,7 | 0,3 | 166 | 244 |
| 26 | JC187-ST-47-48 | 6 | 18 | 0,29 | 0,01 | 4,2 | 0,3 | -26,7 | 0,4 | 163 | 249 |
| 27 | JC187-ST-48-49 | 5 | 20 | 0,29 | 0,02 | 4,3 | 0,4 | -27,0 | 0,3 | 161 | 257 |
| 28 | JC187-ST-49-50 | 6 | 23 | 0,29 | 0,01 | 4,5 | 0,3 | -27,0 | 0,3 | 164 | 230 |
| 29 | JC187-ST-50-51 | 5 | 18 | 0,30 | 0,01 | 4,4 | 0,3 | -27,0 | 0,3 | 168 | 250 |
| 30 | JC187-ST-51-52 | 5 | 21 | 0,31 | 0,01 | 4,4 | 0,3 | -27,0 | 0,3 | 163 | 230 |
| 31 | JC187-ST-52-53 | 5 | 18 | 0,25 | 0,03 | 4,0 | 1,0 | -26,9 | 0,2 | 161 | 235 |
| 32 | JC187-ST-53-54 | 5 | 18 | 0,31 | 0,00 | 4,3 | 0,3 | -26,8 | 0,5 | 165 | 233 |
| 33 | JC187-ST-54-55 | 5 | 18 | 0,31 | 0,02 | 4,3 | 0,2 | -26,8 | 0,4 | 160 | 220 |
| 34 | JC187-ST-55-56 | 5 | 17 | 0,30 | 0,00 | 3,9 | 0,2 | -26,8 | 0,2 | 160 | 241 |
| 35 | JC187-ST-56-57 | 5 | 18 | 0,29 | 0,04 | 4,2 | 0,2 | -26,9 | 0,3 | 158 | 228 |
| 36 | JC187-ST-57-58 | 5 | 19 | 0,30 | 0,07 | 4,1 | 0,3 | -26,8 | 0,4 | 165 | 216 |
| 37 | JC187-ST-58-59 | 6 | 21 | 0,30 | 0,07 | 4,2 | 0,2 | -26,8 | 0,3 | 162 | 245 |
| 38 | JC187-ST-59-60 | 6 | 21 | 0,33 | 0,04 | 4,2 | 0,2 | -26,9 | 0,3 | 183 | 210 |
| 39 | JC187-ST-60-61 | 6 | 22 | 0,29 | 0,02 | 4,3 | 0,3 | -26,9 | 0,3 | 160 | 223 |
| 40 | JC187-ST-61-62 | 6 | 21 | 0,33 | 0,01 | 4,4 | 0,2 | -26,8 | 0,1 | 153 | 223 |
| 41 | JC187-ST-62-63 | 6 | 23 | 0,34 | 0,01 | 4,4 | 0,2 | -26,9 | 0,3 | 165 | 218 |
| 42 | JC187-ST-63-64 | 6 | 25 | 0,31 | 0,04 | 4,3 | 0,2 | -26,9 | 0,3 | 172 | 216 |
| 43 | JC187-ST-64-65 | 6 | 24 | 0,30 | 0,02 | 4,3 | 0,4 | -26,9 | 0,3 | 149 | 278 |



| 44 | JC187-ST-65-66 | 7 | 33 | 0,31 | 0,02 | 4,4 | 0,3 | -27,0 | 0,3 | 162 | 212 |
| 45 | JC187-ST-66-67 | 10 | 50 | 0,29 | 0,04 | 4,3 | 0,1 | -26,9 | 0,4 | 167 | 204 |
| 46 | JC187-ST-67-68 | 7 | 34 | 0,29 | 0,02 | 4,3 | 0,3 | -27,0 | 0,3 | 167 | 211 |
| 47 | JC187-ST-68-69 | 8 | 41 | 0,29 | 0,02 | 4,3 | 0,1 | -26,9 | 0,4 | 159 | 231 |
| 48 | JC187-ST-69-70 | 7 | 35 | 0,31 | 0,01 | 4,5 | 0,3 | -26,9 | 0,3 | 165 | 212 |
| 49 | JC187-ST-70-71 | 10 | 42 | 0,28 | 0,01 | 4,1 | 0,4 | -26,9 | 0,4 | 155 | 221 |
| 50 | JC187-ST-71-72 | 14 | 52 | 0,27 | 0,01 | 4,0 | 0,4 | -26,9 | 0,4 | 150 | 206 |
| 51 | JC187-ST-72-73 | 11 | 54 | 0,24 | 0,01 | 3,6 | 0,2 | -26,8 | 0,4 | 158 | 212 |
| 52 | JC187-ST-73-74 | 7 | 29 | 0,31 | 0,00 | 4,2 | 0,3 | -26,9 | 0,3 | 177 | 203 |
| 53 | JC187-ST-74-75 | 8 | 42 | 0,30 | 0,02 | 4,3 | 0,3 | -27,0 | 0,3 | 160 | 224 |
| 54 | JC187-ST-75-76 | 6 | 26 | 0,31 | 0,02 | 4,5 | 0,3 | -26,8 | 0,2 | 162 | 217 |
| 55 | JC187-ST-76-77 | 6 | 26 | 0,29 | 0,03 | 4,1 | 0,2 | -26,9 | 0,3 | 163 | 214 |
| 56 | JC187-ST-77-78 | 6 | 29 | 0,31 | 0,01 | 4,3 | 0,2 | -26,9 | 0,3 | 172 | 216 |
| 57 | JC187-ST-78-79 | 10 | 46 | 0,31 | 0,03 | 4,3 | 0,1 | -26,9 | 0,3 | 160 | 226 |
| 58 | JC187-ST-79-80 | 18 | 62 | 0,23 | 0,00 | 3,5 | 0,3 | -26,9 | 0,4 | 165 | 198 |
| 59 | JC187-ST-80-81 | 10 | 51 | 0,32 | 0,00 | 4,6 | 0,3 | -26,9 | 0,5 | 162 | 204 |
| 60 | JC187-ST-81-82 | 6 | 30 | 0,32 | 0,01 | 4,3 | 0,6 | -26,7 | 0,4 | 160 | 220 |
| 61 | JC187-ST-82-83 | 6 | 19 | 0,29 | 0,03 | 3,8 | 0,2 | -26,6 | 0,3 | 161 | 211 |
| 62 | JC187-ST-83-84 | 7 | 22 | 0,32 | 0,05 | 4,1 | 0,2 | -26,8 | 0,4 | 161 | 219 |
| 63 | JC187-ST-84-85 | 6 | 22 | 0,32 | 0,01 | 4,2 | 0,2 | -26,9 | 0,3 | 166 | 212 |
| 64 | JC187-ST-85-86 | 7 | 25 | 0,34 | 0,01 | 4,4 | 0,3 | -27,0 | 0,4 | 174 | 209 |
| 65 | JC187-ST-86-87 | 6 | 25 | 0,33 | 0,01 | 4,4 | 0,3 | -27,1 | 0,3 | 171 | 215 |
| 66 | JC187-ST-87-88 | 6 | 24 | 0,32 | 0,01 | 4,4 | 0,3 | -27,1 | 0,4 | 164 | 215 |
| 67 | JC187-ST-88-89 | 6 | 25 | 0,32 | 0,01 | 4,4 | 0,3 | -27,1 | 0,4 | 172 | 239 |
| 68 | JC187-ST-89-90 | 11 | 56 | 0,31 | 0,02 | 4,2 | 0,1 | -27,0 | 0,3 | 167 | 217 |
| 69 | JC187-ST-90-91 | 6 | 29 | 0,30 | 0,02 | 4,4 | 0,3 | -26,9 | 0,4 | 160 | 219 |
| 70 | JC187-ST-91-92 | 6 | 27 | 0,32 | 0,01 | 4,5 | 0,3 | -26,9 | 0,4 | 159 | 302 |
| 71 | JC187-ST-92-93 | 7 | 36 | 0,31 | 0,03 | 4,3 | 0,2 | -26,9 | 0,4 | 165 | 208 |
| 72 | JC187-ST-93-94 | 6 | 30 | 0,30 | 0,03 | 4,4 | 0,2 | -26,9 | 0,3 | 161 | 225 |
| 73 | JC187-ST-94-95 | 15 | 59 | 0,31 | 0,03 | 4,4 | 0,3 | -26,9 | 0,3 | 171 | 208 |
| 74 | JC187-ST-95-96 | 14 | 60 | 0,26 | 0,04 | 4,2 | 0,2 | -26,8 | 0,4 | 156 | 198 |
| 75 | JC187-ST-96-97 | 12 | 59 | 0,11 | 0,05 | 1,8 | 0,1 | -26,4 | 0,5 | 131 | 264 |
| 76 | JC187-ST-97-98 | 14 | 62 | 0,17 | 0,00 | 2,9 | 0,4 | -26,6 | 0,5 | 151 | 218 |
| 77 | JC187-ST-98-99 | 27 | 80 | 0,12 | 0,03 | 2,1 | 0,3 | -26,3 | 0,1 | 134 | 285 |
| 78 | JC187-ST-99-100 | 32 | 77 | 0,15 | 0,01 | 2,4 | 0,1 | -26,6 | 0,5 | 171 | 404 |
| 79 | MC05-0-10cm | 4 | 20 | 0,27 | 0,02 | 4,0 | 0,7 | -26,4 | 0,5 | 154 | 189 |
| 81 | MC05-20-30cm | 9 | 34 | 0,30 | 0,01 | 4,3 | 0,4 | -26,7 | 0,4 | 165 | 162 |
| 83 | MC05-40-50cm | 13 | 55 | 0,26 | 0,01 | 3,7 | 0,1 | -26,8 | 0,3 | 145 | 159 |
| 84 | MC06-40-50cm | 8 | 26 | 0,26 | 0,01 | 3,7 | 0,3 | -26,3 | 0,5 | 140 | 161 |
| 85 | MC06-20-30cm | 6 | 23 | 0,20 | 0,01 | 3,3 | 0,5 | -26,3 | 0,4 | 154 | 168 |
| 86 | MC06-0-10cm | 4 | 19 | 0,25 | 0,02 | 3,5 | 0,5 | -26,4 | 0,4 | 158 | 160 |
| 87 | MC07-top | 8 | 28 | 0,27 | 0,01 | 3,4 | 0,4 | -25,9 | 0,4 | 159 | 162 |
| 88 | MC07-20-30cm | 11 | 40 | 0,29 | 0,01 | 3,5 | 0,4 | -25,9 | 0,5 | 161 | 155 |
| 89 | MC07-40-50cm | 7 | 22 | 0,26 | 0,01 | 3,3 | 0,2 | -25,7 | 0,5 | 146 | 161 |
| 90 | MC03-0-10cm | 10 | 35 | | | 3,9 | 0,3 | -26,5 | 0,7 | 122 | 198 |
| 91 | MC03-40-50cm | 6 | 17 | | | 3,4 | 0,7 | -26,4 | 0,6 | 178 | 198 |
| 92 | MC04-0-10cm | 4 | 22 | | | 4,7 | 0,6 | -27,1 | 0,6 | 154 | 206 |
| 93 | MC04-30-40cm | 3 | 20 | | | 5,1 | 0,4 | -26,9 | 0,6 | 163 | 202 |




**Table 2**. Radiocarbon measurements on 30 selected samples. $F^{14}R$ was calculated using the radiocarbon age of the atmosphere in 2019 ($F^{14}C = 1.019 \pm 0.0004$; Hua et al., 2022).

| Sample code | NEIF reference number | $F^{14}C$ | | $\delta^{13}C$ (‰) | $F^{14}R$ | |
|---|---|---|---|---|---|---|
| | | mean | std | | mean | std |
| JC187-ST-25-26 | SUERC-107302 | 0.920 | 0.004 | -26.693 | 0.902 | 0.004 |
| JC187-ST-30-31 | SUERC-107307 | 0.914 | 0.004 | -26.808 | 0.896 | 0.004 |
| JC187-ST-32-33 | SUERC-107308 | 0.922 | 0.004 | -26.830 | 0.904 | 0.004 |
| JC187-ST-36-37 | SUERC-107309 | 0.925 | 0.004 | -26.745 | 0.907 | 0.004 |
| JC187-ST-40-41 | SUERC-107310 | 0.923 | 0.004 | -26.814 | 0.905 | 0.004 |
| JC187-ST-43-44 | SUERC-107311 | 0.949 | 0.004 | -26.195 | 0.930 | 0.004 |
| JC187-ST-45-46 | SUERC-107312 | 0.913 | 0.004 | -26.539 | 0.895 | 0.004 |
| JC187-ST-57-58 | SUERC-107316 | 0.931 | 0.004 | -26.878 | 0.913 | 0.004 |
| JC187-ST-65-66 | SUERC-107317 | 0.964 | 0.004 | -26.994 | 0.945 | 0.004 |
| JC187-ST-68-69 | SUERC-107318 | 0.952 | 0.004 | -26.807 | 0.934 | 0.004 |
| JC187-ST-72-73 | SUERC-107319 | 0.937 | 0.004 | -26.960 | 0.919 | 0.004 |
| JC187-ST-75-76 | SUERC-107320 | 0.950 | 0.004 | -26.870 | 0.932 | 0.004 |
| JC187-ST-77-78 | SUERC-107321 | 0.950 | 0.004 | -26.893 | 0.932 | 0.004 |
| JC187-ST-79-80 | SUERC-107322 | 0.940 | 0.004 | -26.947 | 0.922 | 0.004 |
| JC187-ST-81-82 | SUERC-107326 | 0.933 | 0.004 | -26.692 | 0.915 | 0.004 |
| JC187-ST-85-86 | SUERC-107327 | 0.938 | 0.004 | -26.975 | 0.920 | 0.004 |
| JC187-ST-89-90 | SUERC-107328 | 0.953 | 0.004 | -27.020 | 0.935 | 0.004 |
| JC187-ST-91-92 | SUERC-107329 | 0.944 | 0.004 | -26.931 | 0.926 | 0.004 |
| JC187-ST-95-96 | SUERC-107330 | 0.912 | 0.004 | -26.701 | 0.894 | 0.004 |
| JC187-ST-96-97 | SUERC-107331 | 0.811 | 0.004 | -26.050 | 0.795 | 0.004 |
| JC187-ST-98-99 | UCIAMS-274854 | 0.846 | 0.007 | | 0.829 | 0.007 |
| JC187-MC05-0-10 | SUERC-108883 | 0.949 | 0.004 | -25.407 | 0.930 | 0.004 |
| JC187-MC05-40-50 | SUERC-108884 | 0.944 | 0.004 | -26.013 | 0.925 | 0.004 |
| JC187-MC06-0-10 | SUERC-108889 | 0.930 | 0.004 | -25.683 | 0.912 | 0.004 |
| JC187-MC07-0-10 | SUERC-108890 | 0.924 | 0.004 | -25.755 | 0.906 | 0.004 |
| JC187-MC07-20-30 | SUERC-108891 | 0.926 | 0.004 | -25.758 | 0.908 | 0.004 |
| JC187-MC03-0-10 | SUERC-108892 | 0.812 | 0.004 | -26.300 | 0.796 | 0.004 |
| JC187-MC03-30-40 | SUERC-108893 | 0.807 | 0.004 | -26.197 | 0.791 | 0.004 |
| JC187-MC04-0-10 | SUERC-108894 | 0.968 | 0.004 | -26.734 | 0.949 | 0.004 |
| JC187-MC04-30-40 | SUERC-108898 | 0.960 | 0.004 | -26.697 | 0.941 | 0.004 |






# 7 Appendices

**Figure A1. X-ray photograph of the sediment trap cut in half and data collected using a Multi-Sensor-Core-Logger on the sediment trap**



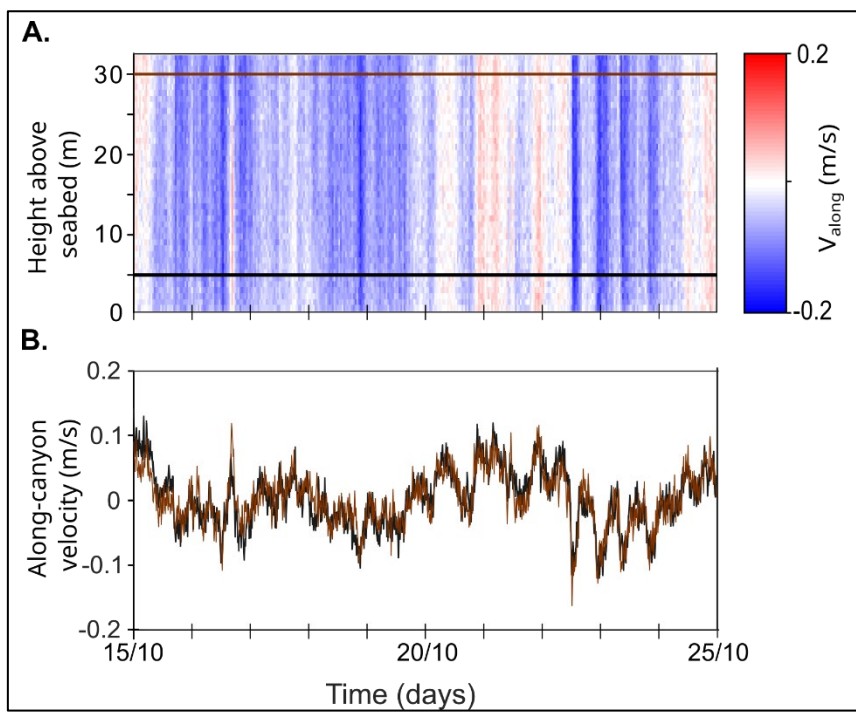

**Figure A2: Velocity along canyon ($V_{along}$) and frequency analysis for the period 15th to 25th October 2019 (see Fig. 2A for context). A.** Time series of $V_{along}$ between 15th October and 25th October 2019. The yellow and black horizontal lines indicate the locations of the arrays displayed in B. **B.** $V_{along}$ speeds at 5 and 30 m above canyon floor.

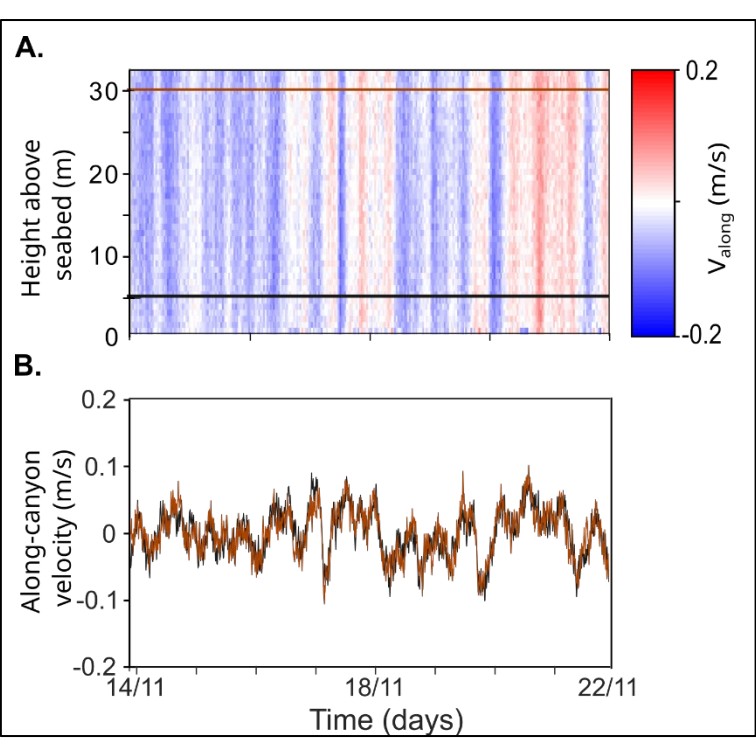




**Figure A3: Velocity along canyon ($V_{along}$) and frequency analysis for the period 14$^{th}$ to 22$^{nd}$ November 2019 (see Fig. 2A for context). A.** Time series of $V_{along}$ between 14$^{th}$ and 22$^{nd}$ November 2019. The yellow and black horizontal lines indicate the locations of the arrays displayed in B.

**Data Availability**

Data associated with the paper have been submitted to an open access repository 'SeaNoe', for which a DOI will be sent.

**Author Contributions**

SH designed the study, analyzed the oceanographic data, performed the measurements for carbon elemental and isotopic compositions, acquired funding for carbon measurements and wrote the manuscript draft. MLB analyzed the sediment trap
(MSCL scanning, X-ray acquisition, description and grain size measurements), extruded samples from the trap, and contributed significantly to the design and writing of the paper. PJT acquired the NERC funding and led the  cruise JC187. NB produced the summary figure. GS analyzed the geochemical data and edited the manuscript. NB, GS, and BD contributed to the design of the manuscript. RGH, VG, FB, CR SS, SA analyzed and commented on the geochemical data. CV and RSJ analyzed and commented on the oceanographic data. All authors reviewed and agreed on the final version of the manuscript.

**Competing interests**

The Authors declare that they have no conflict of interest.

**Acknowledgements**

SH has received funding from the European Union's Horizon 2020 research and innovation programme under the Marie Sklodowska-Curie grant agreement No 899546. MLB was funded by Leverhulme Trust Early Career
Fellowship ECF-2021-566. PJT acknowledges UK Natural Environment Research Council (NERC) grants NE/ R001952/1 and NE/S010068/1. SH and BD acknowledge funding from the INSU LEFE programme for the project "TOC-CC". We thank the captain, the crew and the scientific team of the James Cook cruise JC187. We are grateful to Katie Maier (NIWA) for advice on how to remove sediment from the trap. We also thank Neil Tunstall (Durham University Geography Laboratories) for designing the necessary equipment and methodology for extruding the trap
sediment, and Ed Pope (Durham University) who assisted with removing the trap sediment. Finally we thank



Philippa Ascough (NEIF radiocarbon facility), Caroline Gauthier (LSCE), Florence Savignac (ISTeP) and Loïc Michel (ULiège) for carbon isotope and Rock-Eval analyses.

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
