# Peer review of "How is particulate organic carbon transported through the river-fed Congo Submarine Canyon to the deep-sea?"

_EGUsphere, 2024_

## Referee Comment (RC3)

Review Comment

"How is particulate organic carbon transported through the river-fed Congo Submarine Canyon to the deep-sea?"

By Sophie Hage et al.,

Submitted to Biogeosciences

**General Comments :**

This manuscript describes a study on transport mechanisms of organic carbon throughout the Congo River Canyon towards the deep sea. It combines geophysical (turbidity flow measurements) with geochemical (organic carbon properties) methods to understand processes and organic carbon properties important to understand the fate of terrestrial organic carbon entering the coastal and deep ocean. Understanding transport mechanisms and fate of organic carbon is crucial to estimate its potential for degradation and burial with regards to effects on the global climate. While a lot of studies focus solely on seabed surface sediment samples, this study also examines the suspended sediments collected through a sediment trap, making it an important contribution in the field of organic carbon transport and suitable for publication in *Biogeosciences*. The manuscript has a clearly indicated motivation, well described methodology and logical discussion. Despite minor comments outlined below, I suggest accepting it for publication in *Biogeosciences.*

One of the main points I noticed is the use of particulate organic carbon and total organic carbon being in some sections a bit confusing/inconsistent and used synonymously. While the introduction mentions particulate organic carbon, the results and major part of the discussion does not and suddenly section 5.3 does talk about particulate organic carbon again. It might be good to find a balance between particulate and total organic carbon (where applicable) throughout the whole text to avoid confusion for the reader.

**Specific Comments:**

Line 66 ff:
A question from someone not working with submarine canyons: "This lack of understanding…" – does that make canyons the "perfect" analogue to study and understand these processes? If so, it might be worth mentioning it.

Line 70 ff:

17 May 2024

Reading about particulate organic carbon made me right away wonder if you also compared the data to seabed sediment data, which you did, so it might be worth to consider mentioning this important comparison here already.

Line 73:
"… the Congo River ranks fifth in terms of global particulate organic carbon" – are those the same 7% of organic carbon mentioned in the abstract?

Line 109:
The mentioned number of average annual discharge of 40,000 m$^3$/s for the years 1903-1995, are those the most recent numbers available?

Line 165 ff:
The fourteen mentioned turbidity profiles are they all from M9 mooring?

Line 169:
"…the 14$^{th}$ January event." sound really specific. While it is mentioned in the intro as the strongest event where cables were destroyed, here this is not mentioned again and one might wonder if the expression should be known. Can you either elaborate on that with half a sentence or leave it out?

Section 3:
Grain size analysis description is missing from the method section; "grain size using a Mastersizer laser particle size analyser" (l. 211 f.) is not sufficient as a method description. This becomes even more clear, when looking at the data, divided into D90 and D50, which I have to confess, despite having worked with grain size data before, I do not know what this entails. It might be good to provide this additional piece of information for the reader (including any sample preparation for the grain size measerments).

Line 409:
"…turbidity currents carry organic carbon at a much higher transit flux…". While this makes perfect sense to me in theory, I am wondering how in figure 8 the highest amounts of TOC only match the velocity peak of turbidity current 1, while in the 2-5 interval TOC is elevated during the velocity peak, but the highest TOC values are coinciding with low velocities and a similar pattern can be observed for the 6-8 interval. Can you discuss this a bit further?

Line 415:
"…did not allow sands transported at the base of turbidity currents to be collected." According to your results (p. 42, l. 348), you have sandy samples. Are those the exceptions of small amounts of sands transported above 30 m or is this inconsistent?

Line 423 f:

"…assuming that tides and turbidity currents have the same sediment concentrations." Does this assumption make sense at all? You are proving it wrong below, but it seems striking from the beginning.

Line 431ff:
The paragraph about TOC appears a bit sudden in this section. Maybe consider moving it into the carbon section (5.2)?

Line 452:
"…(at least to the naked eye)." indicates that you did not do any X-ray for those cores, while you did so for the sediment trap. Is there a possibility to still add this data?

Line 457:
Here you state that you estimate the transit flux of organic carbon transported by turbidity currents ten times higher than for tides, while in l. 409 (page before) it says two to three times. Is this a different context and therefore reason for the different numbers?

Line 512:
What does the part "… in the laboratory," mean in this context? Generally, marine organic matter degrades faster than terrestrial organic matter, in the water column, not in the laboratory.

Line 569 f:
"…remove efficiently particulate organic carbon from the terrestrial biosphere to the deep-sea." – What is meant here, is terrestrial originating organic carbon, right? The sentence reads a bit as if the canyon removes it directly from land. I suggest adding a few words for clarification, such as "… and export it to the deep-sea."

**Editorial Comments:**

Line 30:
"Remarkably" appears to me a bit odd to start the sentence. Is it that remarkably?

Line 137 f:
"Piston cores were recently retrieved…" – when is recent? Could you add a year to it?

Line 203 & 211:
It might be more useful to mention the number of multi-cores collected (l. 203), rather than all 91 samples (l. 211).

Line 219:
Where was the first set of samples measured? You mention locations for the latter two sets of samples, but not the first.

17 May 2024

Line 221:
"(ULiege, Belgium)" for consistency.

Line 226 ff:
Consider adjusting the sentences "Knowing the TOC of each sample, we weighed adequate masses [into silver capsules] […]. Samples were acidified with 1M hydrologic acid to remove inorganic carbon." Otherwise, it sounds a bit like the silver capsules are needed for organic carbon removal.

Line 229 ff:
The sentence mentions measurements in triplicates measured at two different locations. Which is the third or where were they measured twice?

Line 241:
Is there a concentration of the hydrochloric acid to be mentioned?

Line 347:
"1.8 and 4.6 %[,] respectively …" is missing a comma.

Line 349:
"TOC [measured] at the base"? "found" does not seem to be the correct word.

Line 426:
"… and far most sediment" should be "far MORE sediment".

Line 428:
"km3" is missing the 3 in superscript.

Line 541:
"…compared to THE Congo RIVER"?

Line 554:
"…rapid erosion rateS"

Figure 1:
- Line 145: "A. Map of THE Congo River…"
- Line 347: Where "cores" are mentioned, are those the piston cores, mentioned in the text? Writing multi-cores and cores, makes one wonder what kind of cores they are.
- A: "RDC" is not explained in the figure caption.
- A: "JC187 Multi-core" looks like the number of a single multi-core, but there are multiple ones listed with different numbers, unless the reason for this number is explained in the caption, I suggest to simply write "multi-core" in the legend.

- A: The superscript number 1 is hard to find in the figure caption, maybe consider writing "(this study)" behind the multi-core and "(Baudin et al., 2017)" behind the lob core and trap, also "Trap" is capitalized, while "core" is not.
- B: Add number "C" into the image next to the profile.

Figure 2:
- Line 189: should Congo Submarine Canyon be capitalized? For consistency.

Figure 3:
- Line 290: the mentioned yellow line is not yellow.
- C: The highlighted frequency of "12.5 h" should be mentioned in the caption. Also, shouldn't it be "Hz" as indicated at the x-axis? cf. l. 303 where it says the same.

Figure 4:
- Can the figure be improved by changing it into three rows? That way (B) will be better visible and comparable to (C).

Figure 5:
- Is there a reason for the bathymetry in the background? To me it seems rather confusing than helpful as it looks like the data continues into the sediment. The first time I looked at the data I was looking for a legend explaining the "additional" data to me, until I realized, it was the bathymetry. If there is not a practical reason for it, maybe consider removing it to avoid confusion or mention it in the caption.

Figure 6:
- What are the blue lines and the highlighted intervals? Information is missing in the caption.
- C: What are the two different types of data?
- E: Is there a reason for the missing lines connecting the black data points?
- Line 340, 341 & 343: l. 340 mentions "trap succession" while both following lines mention "trapped succession", is that correct?

Figure 7:
- Description for panel (B) is missing, (C) is labelled (B) and (D) is missing as a letter before of the description.
- As for fig. 1, it might be helpful to add the references into the figure legend to make it more clear which data is from this study, and which is not.

Figure 8:
- Line 400: "multi-sensor-core-logging" used to be capitalized on page 18 in the text. Adapt for consistency.
- Why are certain turbidity currents grouped? Could you add one explanatory line to the figure caption, including the reason for the highlighted intervals they represent?

Figure 9:
- Line 466: "…with data river data…", one "data" too many.
- B: mentions negative and positive values in the idealized velocity profile. I do not see any values. I am also not sure, I understand what those profiles are showing. Why is the height (water depth?) so different and why does the mooring appear to be at different depths? They might need a bit more explanation.

Table 1:

- It might be clear from the sample code, but I suggest adding some info about sample type into the table (which are the sediment trap and which the seabed samples).
- Is there a reason why the very last four samples in the table do not have TN values?

---

## Author Response (AR1)

Our authors comments (AC) are written in green in response to the reviewers comments (RC)

RC1: 'Comment on egusphere-2024-900', Miquel Canals, 26 Apr 2024

This is an excellent paper by all means, based on a consistent data set resulting from 4 months of in situ monitoring. It proves how efficient, fast and recurrent the transfer of large amounts of river-sourced organic carbon can be. The magnitudes involved are particularly remarkable. So far, the Congo River-canyon-deep sea fan system constitutes a rather unique study case in the world's ocean, which hopefully will pave the way for future research in other similar systems, both present and past.

AC1.1: We thank you for this positive feedback on our work.

Corrections required

1. The caption of Fig. 3 refers to "The yellow and black horizontal lines…" while there is no yellow horizontal line in the figure (but red).

AC1.2: The caption will be changed to "The brown and black horizontal lines…"

2. References need cross-checking. Some references cited in the main text are missing in the References section (e.g. Canals et al., 2006; Ciais et al., 2013, Friedlingstein et al., 2022,….).

AC1.2: The references have been checked and are now complete.

RC2: 'Comment on egusphere-2024-900', Pere Puig, 13 May 2024

Review of Hage et al., How is particulate organic carbon transported through the river-fed Congo Submarine Canyon to the deep-sea? by Pere Puig.

General comment:

This paper presents hydrodynamic and compositional data from moored oceanographic instrumentation and sediment cores to describe and characterize the carbon and sediment transport mechanisms that operate in the Congo Canyon. Even the sediment transport mechanisms in this submarine canyon has been thoroughly addressed in previous papers, particularly the role of turbidity currents, this contribution provides novel data and analyses from the particles collected by a moored sediment trap and from sediment cores, which combined with data from published literature, considerably improves the understanding of carbon transport in the Congo Submarine Canyon.

**AC2.1:** We thank you for this positive feedback on our work.

Overall, the paper is well written, but I found many restatements of the same idea that perhaps could be simply mention once, and also a mixing of contents in various sections. For instance, the Methods includes many details to previous works, which perhaps should be included in the Background section, without restating them, and there are methodological parts included in the Results section. The Methods also includes many references to Figures that are not strictly needed, since they should be introduced later in the manuscript, when it would be more appropriated to follow the reasoning of the paper.

**AC2.2:** We agree with you that some sentences could be deleted and/or streamlined so we will rephrase the manuscript accordingly. In particular, we will delete the first five sentences of sub-section 3.1 (Methods), and the first five sentences of sub-section 4.1 (Results). We also removed all references to figures from the Methods and Conclusions sections. Finally, carbon flux calculations have been moved from Results to Methods.

Another general comment refers to the role of (internal) tides in the transport of particles. In several parts of the manuscript they are considered, along with the turbidity currents, as a sediment transport mechanism contributing to an efficient transfer of river-derived particulate organic carbon from the Congo River to the Congo submarine fan. However, if the residual tidal currents are directed up-canyon, they contribute to retain particles within the canyon interior, rather to promote their transfer towards the submarine fan. This concept should be clarified throughout the paper, as the same misleading statement is mentioned in several parts of the manuscript.

**AC2.3:** We agree with you that the role of tides could be better explained, and that the role of turbidity currents could be better highlighted as turbidity currents are the only process moving material strictly downslope to the deep-sea fan. The following sentences will be modified accordingly:

New Lines 416-418: "Tides have a net residual flow that is oriented up-canyon retaining particles within the canyon before turbidity currents transport particles with a net transit direction oriented downstream towards the deep-sea."

Further, we will rephrase our subsection 5.1 summary as follows:

"In summary, turbidity currents rapidly move sediment and particulate organic carbon downslope in one go at a transit flux that we estimate to be ten times higher compared to the flux induced by tides. It then appears that the fine material flushed episodically by the top part of turbidity currents into the canyon is then kept in suspension and mixed in the canyon water column by tides until a new turbidity current transports particles to the deep-sea. The impact of these combined hydrodynamic processes on organic carbon composition (i.e., isotopic composition and Rock-Eval indexes) is further discussed in the next section."

Finally, we will add "up-canyon transport rates" to a conclusion sentence.

Specific comments:

L 54-56: I do not tend to suggest the citation of my own papers when I provide a review of a submitted manuscript, but in this case, I clearly miss here a reference to the review by Puig et al. 2024 (Annu. Rev. Mar. Sci. 2014. 6:53–77), which specifically address this aspect. The review not only includes the direct measurements during the past decades, but it goes over the published literature since the first instrumented record obtained in a submarine canyon.

**AC2.4:** Thank you for suggesting this paper which is clearly relevant to our work. We will add a reference to the paper in the Introduction

L 61: I am not sure that Shepard et al. 1979 mentioned the mechanisms of cascading and upwelling in their seminal book.

**AC2.5:** We will remove Shepard et al. 1979 from the reference list.

Figure 1A: Change the text "see B", next to the small square, by "Fig. 1B".

**AC2.5:** "see B" will be replaced with "Fig. 1B" in Figure 1A.

Subsection 3.1: This sub-section looks more like a Background section than a Methods section, as most of the presented data is already published. The Methods

should simply describe, aseptically, the methodology used in the paper, so any potential reader could understand what has been done and replicate the same approach in other study areas.

**AC2.6:** We agree with you that the first half of sub-section 3.1 (Methods) were not needed and will thus be deleted, particularly because this part is already stated in the Settings section (2.1).

L 173: There is no need to refer to Fig. 3 here, it would be better to introduce it in the Results section

**AC2.7:** Reference to Fig. 3 will be removed L173

L 175: Is this reference of Hamilton (1847) strictly needed? Besides, it is not included in the bibliography.

**AC2.8:** This citation will be removed from previous L175.

L 176-177: This sentence and the reference to Fig. 4 belongs to Results.

**AC2.9:** This sentence will be removed.

L 185: Are turbidity units correctly expressed, both here and in the axis of Fig. 5D? "$m^{-1}$" corresponds to units of beam attenuation coefficient (when a transmissometer connected to the CTD is used), but "$sr^{-1}$" is new to me. Besides, reporting two different units for the same variable is not common. This should be clarified.

**AC2.10:** Thanks for noticing this, the turbidity units will be corrected to m-1 both in the text and in Fig. 5

L 233: Table 1 includes results. It should be mentioned for the first time in the Results section (when appropriated) and not here.

**AC2.11:** Table 1 will be removed from L233 and will only be referred to in the Results section

L 237: The same for Table 2.

**AC2.12:** Table 2 will be removed from L237 and will only be referred to in the Results section

L 254: The same for the references to Figs. 6 and 7.

**AC2.13:** Figs. 6 and 7 will be removed from L254 and will only be referred to in the Results section.

L 261-262: The same for the references to Fig. 7 and Table 1.

**AC2.14:** Fig. 7 and Table 1 will be removed from L 261-262 and will only be referred to in the Results section

L 275-280: This part of the manuscript is repetitive, and somehow irrelevant for a Results section. Most of this information (and references) could be included in a Background section and the Results section should start straight to the point, avoiding referring to previous works.

**AC2.15:** We agree and we will thus remove L 275-279.

L 290: Change "yellow" by "red"

**AC2.16:** "yellow" will be replaced with "brown"

Figure 3B: Include legend identifying the line color code (similarly to Fig. 3C).

**AC2.17:** "black line" and "brown line" will be added to the figure legend.

L 305-309: Needed?

**AC2.18:** We agree that these sentences were not strictly needed, so they will be removed.

L 318: It is unclear how a sediment trap can "measure" tides and turbidity currents. I assume you refer to their deposit: If so, please rephrase the sentence to avoid confusion.

**AC2.19:** This sentence will be modified as follows:

"Sedimentary deposits associated with tides and turbidity currents measured by the ADCP are recorded in the sediment trap deployed just below the ADCP instrument"

L 323: Restated information in L 319. Decide where to include the reference of the height above the seabed, here or there, since both sentences are pretty close.

**AC2.20:** We will delete the first instance of this information from L323.

Figure 5D: Check the used turbidity units.

**AC2.21:** Turbidity units will be changed to m-1

Figure 6: Overview of the data collected from the sediment trap… and the ADCP? Plot A does not belong to the sediment trap.

**AC2.22:** We will add "from the ADCP and the sediment trap" to Figure 6 title.

Figure 7: Descriptions form plot B is missing in the figure caption, and the text after B. corresponds to the content in plot C. The letter in bold used as reference previous to the text corresponding to plot D should be also included in the figure caption, since it is missing.

**AC2.23:** Thanks for noticing those mistakes, the caption will be modified accordingly:

"Measurements made on trap and seabed samples collected in the Congo Canyon and Channel, with previous data from Congo River and lobe. **A.** Total organic carbon content (TOC) against grain size D90 in micrometres. **B.** Relative 14C enrichment (relative to year of sample collection in 2019; F14R) against D90 in micrometres. **C.** Relative 14C enrichment (relative to year of sample collection in 2019; F14R) against carbon stable isotope ratios ($\delta$13C). River data are taken from Spencer et al. (2012) and Hemingway et al. (2017) and are averaged between samples collected in one year (labelled 2008, 2011, 2012 and 2013). **D.** Hydrogen index against oxygen index derived from Rock-Eval pyrolysis/oxidation. Lobe data are taken from Baudin et al. (2017). "

Subsection 4.3: This subsection (until L 389) belongs to Methods.

**AC2.24:** This subsection will be moved to the Methods section (new sub-section 3.4)

L 375: There is no need to refer to Fig. 8 in this methodological description.

**AC2.25:** Reference to Fig. 8 will be removed from this description.

L 405: The residual tidal flow goes up-canyon, and therefore, they retain particles in their transit from the River to the deep-sea fan, rather than transporting them.

**AC2.26:** We will add a new sentence specifying this aspect to the Result section (4.3):

"Tides have a net residual flow that is oriented up-canyon retaining particles within the canyon before turbidity currents transport particles with a net transit direction oriented downstream towards the deep-sea."

We will also replace the word "transport" with "fate" at L405

Finally, we will rephrase our subsection 5.1 summary as follows:

"In summary, turbidity currents rapidly move sediment and particulate organic carbon downslope in one go at a transit flux that we estimate to be ten times higher compared to the flux induced by tidesIt then appears that the fine material flushed

episodically by the top part of turbidity currents into the canyon is then kept in suspension and mixed in the canyon water column by tides until a new turbidity current transports particles to the deep-sea. The impact of these combined hydrodynamic processes on organic carbon composition (i.e., isotopic composition and Rock-Eval indexes) is further discussed in the next section."

L 423-424: You cannot assume that tides and turbidity currents have the same sediment concentration. As it is mentioned afterwards, turbidity currents are much more concentrated! So it is better not to compute and compare fluxes in this way, as it can induce confusion to the reader.

AC2.27: We agree and we will tone this down by deleting L420-424. We will state that turbidity currents have a transit flux at least 6 times higher compared to the flux induced by tides, based on our estimated fluxes from Fig. 8 only.

Figure 9: Very nice summary figure.

AC2.28: Thank you for this positive comment.

L 521-522: The list of submarine canyons included in the review by Puig et al (2014) includes more than few of them.

AC2.29: Thank you for this relevant suggestion, we will add a reference to Puig et al. (2014) at L521-522.

L 582: Up-canyon transport rates?

AC2.30: "up-canyon" will be added to previous L582.

L584-586: Avoid using references in the Conclusions, as they should simply reflect the outputs of the paper. Such type of referencing should be included in the Discussion section.

AC2.31: References will be removed from the conclusions.

Bibliography: There are missing references, it needs to be revised.

AC2.32: The reference list was checked and will be complete in the updated version of our manuscript.

Sincerely,

Pere Puig

**In green are our author comments (AC) in response to Reviewer 3's comments**

**RC3: Review Comment:** "How is particulate organic carbon transported through the river-fed Congo Submarine Canyon to the deep-sea?" 17 May 2024

Comments : This manuscript describes a study on transport mechanisms of organic carbon throughout the Congo River Canyon towards the deep sea. It combines geophysical (turbidity flow measurements) with geochemical (organic carbon properties) methods to understand processes and organic carbon properties important to understand the fate of terrestrial organic carbon entering the coastal and deep ocean. Understanding transport mechanisms and fate of organic carbon is crucial to estimate its potential for degradation and burial with regards to effects on the global climate. While a lot of studies focus solely on seabed surface sediment samples, this study also examines the suspended sediments collected through a sediment trap, making it an important contribution in the field of organic carbon transport and suitable for publication in Biogeosciences. The manuscript has a clearly indicated motivation, well described methodology and logical discussion. Despite minor comments outlined below, I suggest accepting it for publication in Biogeosciences.

**AC3.1:** We thank you for this clear summary and positive feedback on our work.

One of the main points I noticed is the use of particulate organic carbon and total organic carbon being in some sections a bit confusing/inconsistent and used synonymously. While the introduction mentions particulate organic carbon, the results and major part of the discussion does not and suddenly section 5.3 does talk about particulate organic carbon again. It might be good to find a balance between particulate and total organic carbon (where applicable) throughout the whole text to avoid confusion for the reader.

**AC3.2:** We agree that particulate organic carbon (POC) and total organic carbon (TOC) need to be distinguished and we will make this clear in the updated version of our manuscript.

Specific Comments:

Line 66: A question from someone not working with submarine canyons: "This lack of understanding..." – does that make canyons the "perfect" analogue to study and understand these processes? If so, it might be worth mentioning it.

**AC3.3:** Submarine canyons provide the main pathway for particles to travel between continental shelves and the deep-sea, so yes submarine canyons are definitely worth studying when it comes to understanding terrestrial transfer of particles to the deep-sea. We will add the following sentence to the Introduction (Line 46-47):

"Submarine canyons are ubiquitous on the ocean floor, where more than 5000 canyons have been mapped (Harris and Whiteway, 2011)."

Line 70: Reading about particulate organic carbon made me right away wonder if you also compared the data to seabed sediment data, which you did, so it might be worth to consider mentioning this important comparison here already.

AC3.4: We agree with this suggestion and we will modify Line 70 accordingly:

"...we use direct observations of transport and sampling of particulate organic carbon from the Congo Submarine Canyon **water column and seabed**".

Line 73: "... the Congo River ranks fifth in terms of global particulate organic carbon" – are those the same 7% of organic carbon mentioned in the abstract?

AC3.5: No, the 7% correspond to total (i.e. both dissolved and particulate) organic carbon whereas the Congo River ranks fifth in terms of particulate organic carbon globally. We will make this clearer by replacing "total" with "**dissolved and particulate**" in the abstract sentence.

Line 109: The mentioned number of average annual discharge of 40,000 m3/s for the years 1903-1995, are those the most recent numbers available?

AC3.6: Thanks for noting this, we have found a more recent number of 40,500 $m^3$/s for the period 1903-2020 in a study by Laraque et al. 2022, Hydroscience Journal). This number and reference will be added to the updated version of our manuscript.

Line 165: The fourteen mentioned turbidity profiles are they all from M9 mooring?

AC3.7: Yes, all 14 turbidity currents were detected at Mooring M9 and this will be specified in the updated version of the manuscript as follows:

"In this study, we focus on the first eight turbidity currents (between 15th September and 10th December 2019) out of the fourteen events detected in the full time series recorded at mooring M9."

Line 169: "...the 14th January event." sound really specific. While it is mentioned in the intro as the strongest event where cables were destroyed, here this is not mentioned again and one might wonder if the expression should be known. Can you either elaborate on that with half a sentence or leave it out?

AC3.8: We will delete the part on "the 14th January" event in this section as it is already mentioned in the Introduction and it is not further needed in our Methods.

Section 3: Grain size analysis description is missing from the method section; "grain size using a Mastersizer laser particle size analyser" (l. 211 f.) is not sufficient as a

method description. This becomes even more clear, when looking at the data, divided into D90 and D50, which I have to confess, despite having worked with grain size data before, I do not know what this entails. It might be good to provide this additional piece of information for the reader (including any sample preparation for the grain size measerments).

**AC3.9:** We thank you for this suggestion and we agree that details on Grain Size measurements were missing. We will add the following paragraph to Section 3.2:

"All ninety-one samples collected from the trap and the multi-cores were analysed for grain size using Beckman Coulter LS 13 320 Laser Diffraction Particle Size Analyser at the Department of Geography, Durham University. 20 mL of 20% hydrogen peroxide was added to ~0.5 g of sediment sample to remove organics before the sample was centrifuged to remove the supernatant. Samples were then mixed with 20 mL of deionized water and 2 mL of sodium hexametaphosphate solution to limit flocculation. Samples were run through the analyser three times; the runs were compared and if similar then the results were averaged. The D50 (i.e., median grain size) and the D90 (i.e., the particle diameter where 90% of the distribution has a smaller particle size) were computed and are presented in this study."

Line 409: "...turbidity currents carry organic carbon at a much higher transit flux...". While this makes perfect sense to me in theory, I am wondering how in figure 8 the highest amounts of TOC only match the velocity peak of turbidity current 1, while in the 2-5 interval TOC is elevated during the velocity peak, but the highest TOC values are coinciding with low velocities and a similar pattern can be observed for the 6-8 interval. Can you discuss this a bit further?

**AC3.10:** The highest amount in carbon flux is also delayed compared to the max velocity peak in turbidity current 1. All turbidity currents actually show lower TOC contents in the sands (TOC = ~2%) transported in the early stages of turbidity currents (i.e., velocity peaks), compared to muddy material (TOC = ~4.5 %) transported later in the currents and thus deposited later in the trap. So grain size sorting determines TOC contents, impacting final carbon fluxes shown in Figure 8C. However, we expect higher carbon fluxes, due to higher TOC contents in the sands transported during the early stages of the turbidity currents but these TOC-rich sands are not visible in the trap, as explained at Lines 415 and 416, see our response AC3.11.  We will modify Section 5.1 to explain these aspects:

"(...)Despite this, the particulate organic carbon transit flux due to turbidity currents is likely underestimated because the height of our sediment trap (30 m above seabed) did not allow coarse sands transported at the base of turbidity currents to be collected. The base of turbidity currents (e.g., 0 to 5 m above seabed), are usually characterized by the flow's highest velocities and sediment concentrations (Kneller and Buckee, 2000, Sequeiros et al., 2009). These higher velocities can transport larger fresh vegetation debris associated with coarser sands (McArthur et al., 2016).

For example, event beds containing large amount of vegetation debris (TOC of up to 11.3 %) associated with medium sand (in a muddy matrix) were reported in some piston cores collected in the Congo Canyon (Baker et al., in press). Furthermore, vegetation-rich sandy turbidites have been observed in submarine canyon deposits elsewhere (e.g., Saller et al., 2006, Lee et al., 2019, Hage et al., 2020), supporting our hypothesis."

Line 415: "…did not allow sands transported at the base of turbidity currents to be collected." According to your results (p. 42, l. 348), you have sandy samples. Are those the exceptions of small amounts of sands transported above 30 m or is this inconsistent?

**AC3.11:** You are correct, we have sand-sized particles in our trap. However we expect even more and coarser sand particles at the base of turbidity currents which could ne be captured in our 30-m high trap. This is based on the presence of large amount of coarser sands in piston cores collected in the Congo Canyon presented in Baker et al (in press, Geology), as stated in Line 418. We will clarify this aspect in section 5.1, see our response AC3.10.

Line 423 f: "…assuming that tides and turbidity currents have the same sediment concentrations." Does this assumption make sense at all? You are proving it wrong below, but it seems striking from the beginning.

**AC3.12:** We agree this was not clear, as also pointed out by Reviewer 2. We will remove this sentence in the updated version of our manuscript.

Line 431: The paragraph about TOC appears a bit sudden in this section. Maybe consider moving it into the carbon section (5.2)?

**AC3.13:** We agree with you that these sentences about TOC are not needed here, and they are redundant with our result section 4.2. We will delete Lines 431 to 434 in the updated version of the manuscript.

Line 452: "…(at least to the naked eye)." indicates that you did not do any X-ray for those cores, while you did so for the sediment trap. Is there a possibility to still add this data?

**AC3.14:** The multi-cores were not x-rayed and there will be no possibility to x-ray these cores within the timeframe of this paper/project.

Line 457: Here you state that you estimate the transit flux of organic carbon transported by turbidity currents ten times higher than for tides, while in l. 409 (page before) it says two to three times. Is this a different context and therefore reason for the different numbers?

**AC3.15:** We toned down our estimates of the carbon transit fluxes to "at least three to six times", based solely on our Figure 8.

Line 512: What does the part "… in the laboratory," mean in this context? Generally, marine organic matter degrades faster than terrestrial organic matter, in the water column, not in the laboratory.

**AC3.16:** We agree that "in the laboratory" was not relevant in this sentence so we will remove it in the updated version of the manuscript.

Line 569 f: "…remove efficiently particulate organic carbon from the terrestrial biosphere to the deepsea." – What is meant here, is terrestrial originating organic carbon, right? The sentence reads a bit as if the canyon removes it directly from land. I suggest adding a few words for clarification, such as "… and export it to the deep-sea."

**AC3.17:** this sentence will be modified as follows:

"Overall, our observations of particulate organic carbon transport in the Congo Submarine Canyon reveal that strong hydrodynamic processes in canyons appear to efficiently transport terrestrial particulate organic carbon and export it to the deep-sea"

**Editorial Comments:**

Line 30: "Remarkably" appears to me a bit odd to start the sentence. Is it that remarkably?

**AC3.18:** "Remarkably" will be deleted

Line 137 f: "Piston cores were recently retrieved..." – when is recent? Could you add a year to it?

**AC3.19:** Piston cores were retrieved in 2019 and this will be added to Line 137.

Line 203 & 211: It might be more useful to mention the number of multi-cores collected (l. 203), rather than all 91 samples (l. 211).

**AC3.19:** "All 91 samples from the trap and the 3 multi-cores" will be specified at Line 203

Line 219: Where was the first set of samples measured? You mention locations for the latter two sets of samples, but not the first.

**AC3.20:** The first set of samples was measured at the Panoply platform (Paris-Saclay), and this will be added.

Line 221: "(ULiege, Belgium)" for consistency.

**AC3.21:** This will be modified accordingly

Line 226: Consider adjusting the sentences "Knowing the TOC of each sample, we weighed adequate masses [into silver capsules] [...]. Samples were acidified with 1M hydrologic acid to remove inorganic carbon." Otherwise, it sounds a bit like the silver capsules are needed for organic carbon removal.

**AC3.22:** This will be modified as suggested

Line 229: The sentence mentions measurements in triplicates measured at two different locations. Which is the third or where were they measured twice?

**AC3.23:** Samples were measured twice at the Panoply platform (Paris-Saclay), and we will make this clearer together with our response AC3.20

Line 241: Is there a concentration of the hydrochloric acid to be mentioned?

**AC3.24:** The NEIF facility used a 37.3% concentrated (i.e., undiluted) HCl and this will be added to the sentence.

Line 347: "1.8 and 4.6 %[,] respectively …" is missing a comma.

**AC3.25:** A comma will be added

Line 349: "TOC [measured] at the base"? "found" does not seem to be the correct word.

**AC3.26:** "found" will be replaced by "measured"

Line 426: "… and far most sediment" should be "far MORE sediment".

**AC3.27:** "most" will be replaced by "more"

Line 428: "km3" is missing the 3 in superscript.

**AC3.28:** A superscript will be added

Line 541: "…compared to THE Congo RIVER"?

**AC3.29:** we will modify as follows: "compared to the Congo Canyon"

Line 554: "…rapid erosion rateS"

**AC3.30:** An "s" will be added

Figure 1:

    - Line 145: "A. Map of THE Congo River…"

**AC3.30:** "the" will be added

    - Line 347: Where "cores" are mentioned, are those the piston cores, mentioned in the text? Writing multi-cores and cores, makes one wonder what kind of cores they are.

**AC3.31:** These correspond to the piston cores studied by Baudin et al. (2017). We will add "piston" to both the caption and the text when referring to piston cores.

    - A: "RDC" is not explained in the figure caption.

**AC3.32:** RDC denotes the "Democratic Republic of Congo" in French. This will be added to the caption.

    - A: "JC187 Multi-core" looks like the number of a single multi-core, but there are multiple ones listed with different numbers, unless the reason for this number is explained in the caption, I suggest to simply write "multi-core" in the legend.

**AC3.33:** JC187 corresponds to the cruise number during which the multi-cores presented in this study were retrieved. This will be added to the legend.

- A: The superscript number 1 is hard to find in the figure caption, maybe consider writing "(this study)" behind the multi-core and "(Baudin et al., 2017)" behind the lob core and trap, also "Trap" is capitalized, while "core" is not.

**AC3.34:** "(this study)" and "(Baudin et al., 2017)" will be added. "Trap" will be replaced with "trap"

- B: Add number "C" into the image next to the profile.

**AC3.35:** "C" will be added

Figure 2:

- Line 189: should Congo Submarine Canyon be capitalized? For consistency.

**AC3.36:** "Congo submarine canyon" will be capitalized

Figure 3:

- Line 290: the mentioned yellow line is not yellow.

**AC3.37:** "Yellow" will be replaced with "brown"

- C: The highlighted frequency of "12.5 h" should be mentioned in the caption. Also, shouldn't it be "Hz" as indicated at the x-axis? cf. l. 303 where it says the same.

**AC3.38:** The frequency (in Hz) corresponds to 1/time, so 12.5h corresponds to 1/Frequency presented in the x-axis. This will be specified in the caption.

Figure 4:

- Can the figure be improved by changing it into three rows? That way (B) will be better visible and comparable to (C).

**AC3.39:** We would prefer to keep the figure as it is, as with the current display, part B is scaled with part C (in terms of X and Y axes dimensions) so that both graphs are comparable.

Figure 5:

- Is there a reason for the bathymetry in the background? To me it seems rather confusing than helpful as it looks like the data continues into the sediment.

The first time I looked at the data I was looking for a legend explaining the "additional" data to me, until I realized, it was the bathymetry. If there is not a practical reason for it, maybe consider removing it to avoid confusion or mention it in the caption.

**AC3.40:** We believe that the bathymetry in the background of this figure is useful because it shows the variations in parameters according to the canyon morphology, as discussed in the text. We will increase the transparency of the bathymetric profile to make it clearer and we will add a note about this in the caption, as follows:.

"The gey line shows a cross section of the across-canyon morphology at the mooring location."

Figure 6:

    - What are the blue lines and the highlighted intervals? Information is missing in the caption.

**AC3.41:** The blue lines will be deleted. The highlighted yellow intervals correspond to periods when turbidity currents are active. This will be specified in the figure caption as follows:

"The yellow rectangles in the background highlight periods when turbidity currents are active."

    - C: What are the two different types of data?

**AC3.42:** The following sentences will be added to Figure. 6 part C caption: "The black line represents the grain size median (D50). The grey line represents the grain size D90."

    - E: Is there a reason for the missing lines connecting the black data points?

**AC3.43:** We only have 21 data points for the F14R measurements so we think that connecting these points would be misleading.

    - Line 340, 341 & 343: l. 340 mentions "trap succession" while both following lines mention "trapped succession", is that correct?

**AC3.44:** "Trapped succession" is more correct and this will be adjusted accordingly.

Figure 7:

    - Description for panel (B) is missing, (C) is labelled (B) and (D) is missing as a letter before of the description.

**AC3.45:** "D." will be added to the caption and a description for panel B and C will be added as follows: "B. Relative 14C enrichment (relative to year of sample collection in 2019; F14R) against D90 in micrometres. C. Relative 14C enrichment (relative to year of sample collection in 2019; F14R) against carbon stable isotope ratios (δ13C)."

    - As for fig. 1, it might be helpful to add the references into the figure legend to make it more clear which data is from this study, and which is not.

**AC3.46:** A reference to the previous studies will be added to the figure legend

Figure 8:

    - Line 400: "multi-sensor-core-logging" used to be capitalized on page 18 in the text. Adapt for consistency.

**AC3.47:** "multi-sensor-core-logging" will be capitalized in the caption

    - Why are certain turbidity currents grouped? Could you add one explanatory line to the figure caption, including the reason for the highlighted intervals they represent?

**AC3.48:** Turbidity currents 2 to 5 and 6 to 8 are so close in time so that we highlighted them together respectively, for figure clarity, and consistency with figure 6. We will add the following sentence to the figure caption:

"The yellow rectangles in the background highlight periods when turbidity currents are active."

Figure 9:

    - Line 466: "…with data river data…", one "data" too many.

**AC3.49:** the first occurrence of "data" will be removed from this sentence caption

    - B: mentions negative and positive values in the idealized velocity profile. I do not see any values. I am also not sure, I understand what those profiles are showing. Why is the height (water depth?) so different and why does the mooring appear to be at different depths? They might need a bit more explanation.

**AC3.50:** The negative and positive values represent the velocities that are directed up-canyon (negative values) and down-canyon (positive values), as observed at the moored ADCP reported in Fig. 3A and 3B. This will be added to the figure caption as follows:

"idealized velocity profile showing the oscillations with a downslope (positive values) and upslope direction (negative values) as observed in Fig. 3A and 3B"

The height is different between B and C because tides influence the water column much higher compared to turbidity currents which are restricted to the lowermost parts of the canyon. Despite the difference in Y-axis scale, the mooring is located at 30 m above seabed in both B and C. The height of the canyon trap is provided in the figure legend "Canyon trap (30m)".

Table 1:

     - It might be clear from the sample code, but I suggest adding some info about sample type into the table (which are the sediment trap and which the seabed samples).

**AC3.51:** A column "sample type" will be added to Table 1

     - Is there a reason why the very last four samples in the table do not have TN values

**AC3.52:** Those last four samples were only measured in ULiège (Belgium) with the triplicate measurements for other samples and N could not be measured in this lab. Instead, those last four samples were measured in duplicates for TOC in ULiège.